evolution/developmental biology

critical size, larval growth, adult body size, adaptive bailout, accelerated pre-adult development

**Author for correspondence:**
Mallikarjun N. Shakarad
e-mail: beelab.ms@gmail.com

# Evolution of reduced minimum critical size as a response to selection for rapid pre-adult development in *Drosophila melanogaster*

Khushboo Sharma, Nalini Mishra and
Mallikarjun N. Shakarad

Evolutionary Biology Laboratory, Department of Zoology, University of Delhi, New Delhi, Delhi 110007, India

(iD) KS, 0000-0003-2278-3635; MNS, 0000-0003-1134-3001

Adult body size in holometabolus insects is directly proportional to the time spent during the larval period. The larval duration can be divided into two parts: (i) pre-critical duration—time required to attain a critical size/critical weight that would result in successful completion of development and metamorphosis even under non-availability of nutrition beyond the time of attainment of critical size, and (ii) post-critical duration—the time duration from the attainment of critical size till pupation. It is of interest to decipher the relative contribution of the two larval growth phases (from the hatching of the egg to the attainment of critical size, and from the attainment of critical size to pupation) to the final adult size. Many studies using *Drosophila melanogaster* have shown that selecting populations for faster development results in the emergence of small adults. Some of these studies have indirectly reported the evolution of smaller critical size. Using two kinds of *D. melanogaster* populations, one of which is selected for faster/ accelerated pre-adult development and the other their ancestral control, we demonstrate that the final adult size is determined by the time spent as larvae post the attainment of critical size despite having increased growth rate during the second larval instar. Our populations under selection for faster pre-adult development are exhibiting adaptive bailout due to intrinsic food limitation as against extrinsic food limitation in the yellow dung fly.

# 1. Background

Holometabolous insect species are characterized by two distinct phases in their life cycle *viz.*, the pre-adult phase which consists of (i) larval and (ii) pupal stages; and adult phase. During the larval life, the energy required for metamorphosis from larval to adult tissue and for the early adult life is accumulated [1–3]. Further, the duration of the larval stage determines the final adult body size. Contrary to common belief, unrestricted growth occurs even at the time of moulting due to the presence of unsclerotized body surface [4]. However, the timing of metamorphosis imposes a restriction on larval duration which directly affects the final adult size and associated life-history traits [5,6]. Insects exhibit varied mechanisms of final body size assessment as a prerequisite for metamorphosis initiation. In holometabolous and some hemimetabolous insects, the process of initiation of metamorphosis is dependent on attaining a certain minimum threshold size called critical size [7–10] beyond which starvation does not alter the time course to metamorphosis [10,11–17]. In *Drosophila*, the size at which 50% of starved larvae successfully eclose as adults (the minimum viable weight for eclosion) is used as a proxy for critical size [16,18]. We use this proxy for critical size/weight in this study.

Critical size being an essential checkpoint during the larval stage acts as a developmental switch for the irreversible process of metamorphosis [13,14,17]. The early phase of larval life consists of the exponential growth phase that ends in the attainment of critical size, while the later, post-critical, phase is marked by linear growth period on the arithmetic scale [19]. In *Drosophila* sp., the final adult body size is determined during this post-critical phase. Thus, the larval duration is split into (i) pre-critical duration, which is defined as the development time spent in attaining the minimum size necessary to complete metamorphosis and emerge as an adult [10], and (ii) post-critical duration, during which additional energy required for maximizing Darwinian fitness is acquired [1,15]. Once critical size is attained, variable size controlling mechanisms operate in different species before they undergo metamorphosis, indicating that these species have unique modes of determining the body size with critical size at its core [4,9,19]. For example, in *Manduca sexta*—a Lepidopteran holometabolous insect—larvae between fourth instar and fifth instar stage whose head capsule size was greater than 5.4 mm were able to successfully pupate else undergo additional moult to sixth instar, thus the data suggest that larvae monitor their size by the growth of head capsule [20]; while in *Oncopeltus fasciatus*—a Hemipteran hemimetabolous insect—larval growth and its size are estimated by abdominal stretch receptors [9]. Critical size is suggested to evolve in response to environmental conditions. For example, in *Drosophila* genus, large-sized species like *D. repleta* have higher critical size—which is larger than the final larval size of small-sized *D. willistoni*. Critical size change, thus, can be one of the drivers of adult body size evolution [18].

*Drosophila melanogaster* is known to occupy ephemeral habitat with limited food and high density and thus is believed to be under strong selection for faster pre-adult development [21]. Previously, it has been reported that *Drosophila* populations under conscious selection for shorter pre-adult duration have reduced body size [22–25]. It has been speculated that critical size might reduce if exposed to conscious selection for accelerated pre-adult development [24]. However, every extant species should have evolved a species-specific critical size that has been optimized over the course of evolution, as the critical size is crucial to survival itself. Previous studies have demonstrated the evolution of critical size in *Drosophila melanogaster* populations under conditions of malnutrition [25], and selection for body size [26], thus exhibiting genetic variability for the trait [11]. For example, under direct artificial selection for change in body size, there is a reduction in the critical size [10,12], while in another study, populations under nutritional stress evolved smaller critical size [25]. In this study, we test the hypothesis that selection for faster pre-adult development reduces the critical size in *D. melanogaster* [24].

We used six populations of *D. melanogaster*, of which three were ancestral controls maintained on a 21-day egg-to-egg discrete generation cycle and three were simultaneously selected for faster pre-adult development and extended reproductive lifespan. The control and selected populations had been through 232 and 126 generations, respectively, at the time of initiation of these experiments. We first assessed the pre- and post-critical duration and critical size in the control and selected populations. Then we evaluated the impact of non-availability of food on the post-critical larval duration, pupal duration and adult body size in control and selected populations. Further, we assessed the impact of selection on larval growth rate. We found that the selected populations have evolved a significantly reduced pre- and post-critical duration and smaller critical size as a correlated response to selection for faster pre-adult development. Interestingly, the selected populations have higher growth rate during the second larval instar, suggesting that they might have preponed their growth owing to a very short post-critical duration.

**Table 1.** Diet composition: standard media (SM) and liquid standard media (LSM). LSM has all ingredients except agar-agar.

| diet composition (1 l) | standard media (SM) | liquid standard media (LSM) |
| --- | --- | --- |
| water | 1180 ml | 1180 ml |
| banana | 205 g | 205 g |
| jaggery | 35 g | 35 g |
| barley flour | 25 g | 25 g |
| yeast | 36 g | 36 g |
| methyl paraben | 2.4 g | 2.4 g |
| ethanol | 45 ml | 45 ml |
| agar-agar | 12.5 g | Zero |

# 2. Methods

## 2.1. Fly husbandry

A total of six *D. melanogaster* populations were used in this study. Of the six populations, three were ancestral controls (Joshi baseline (JB) populations) maintained on a 21-day egg-to-egg discrete generation cycle [27]. The other three were simultaneously selected for faster pre-adult development and extended adult reproductive lifespan [28]. All the six populations were maintained as large outbred populations in Power Scientific Inc., USA environmental chambers/incubators under standard laboratory conditions (SLC) of $25 \pm 2°C$ temperature, $70 \pm 5\%$ RH (relative humidity) and $24 : 0\,L : D$ (light : dark) cycle. The pre-adult stages were reared in glass vials ($9.5 \times 2.3$ cm) with 6 ml standard media (SM) (table 1), whereas the adults were reared in Plexiglas cages ($25 \times 20 \times 15$ cm). The pre-adults were on a single meal of SM in glass vials till emergence as adults, while the adults (in Plexiglas cages) were provided fresh SM every alternate day. All population cages were provided with yeast and acetic acid supplement along with fresh SM 3 days prior to collection of eggs for starting the generation. Each of the control populations was generated in 40 vials, with 50–60 eggs in 6 ml SM per vial and incubated at SLC for 12 days in vials. All the emerging adults of a given population were transferred to a clean, sterile pre-labelled population cage with a fresh plate of SM (figure 1a).

Selected (FLJ- Faster developing, Late reproducing and JB derived) populations were derived from corresponding ancestral controls (JBs) by transferring 60–80 eggs into 6 ml SM vials under SLC. Egg density was kept low so as to avoid larval crowding [29], and the difference in the egg densities of control and selected populations is marginal and is unlikely to differentially affect any traits in the two population types. A total of 160 vials per replicate population were set up. Only the first 15–20 flies emerging from each of 160 vials were transferred to pre-labelled clean breeding cages through the process of 2-hourly vigil checks. The initial population size of each of the selected populations was 2400–3200 individuals. In order to avoid crowding during the adult stage, the emerging adult flies were maintained in two sister cages, with each cage housing adults from 80 vials. Eggs for initiating the subsequent generation were collected after 50% adult mortality was noted in either of the cages, thus ensuring a breeding population size of approximately 1600 flies (numbers similar to control populations) at the time of egg collection. The eggs from the two sister cages were mixed and redistributed into 160 vials to avoid independent evolutionary trajectories in the two sister cages (figure 1b).

## 2.2. Generation of flies for experiments

In order to remove any non-genetic parental effects, eggs were collected from both sets of populations—selected (FLJ) and control (JB) populations, and reared under similar conditions, wherein the selection criteria were relaxed in the selected populations prior to experimentation [22,30]. Eggs were collected on a sterile media plate and exact counts of 50 eggs were dispensed into vials with 6 ml of SM. Forty such vials were maintained per population. All the flies that emerged at the end of 10 (selected) and 12 (control) days from the 40 vials were transferred to pre-labelled clean Plexiglas population cages and provided with uncontaminated SM plate. These flies are henceforth referred to as 'standardized flies'. In the experiments that required a large number of eggs, two sister cages of standardized flies

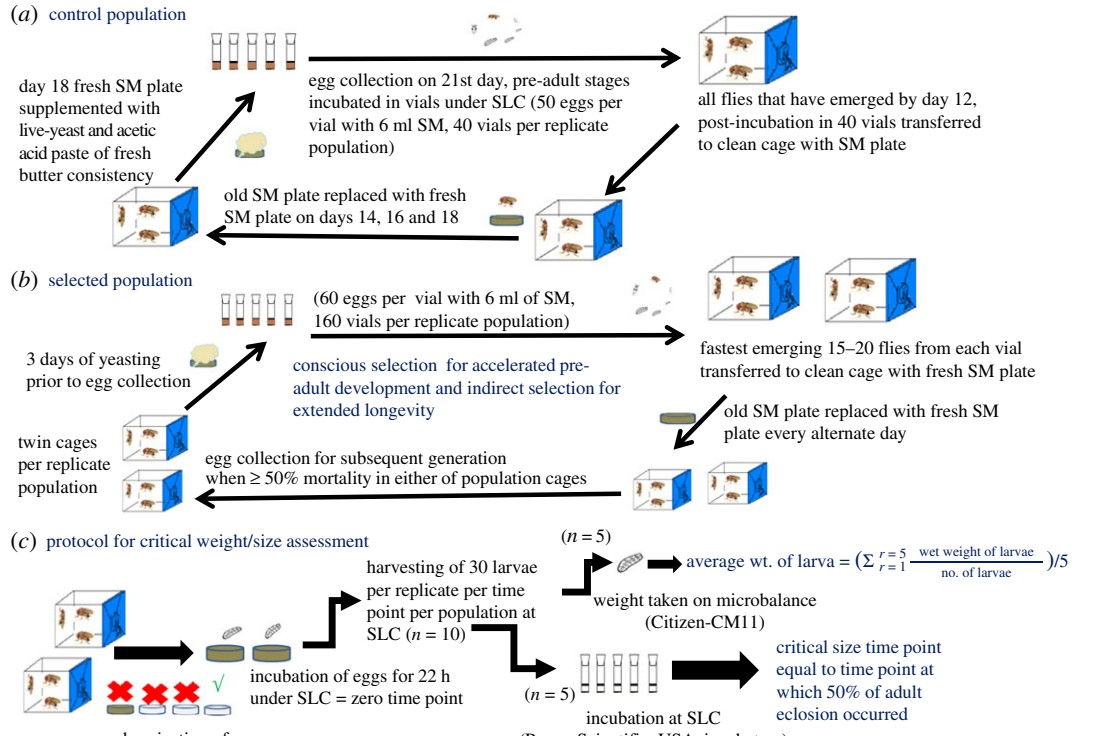

**Figure 1.** Schematic of populations and sampling of larvae: (*a*) control, (*b*) selected populations and (*c*) protocol for sampling of larvae for the assessment of critical size and critical duration in selected and control populations.

were generated by incubating 80 vials of 50 eggs each, per population. Though selected populations were maintained at 60–80 eggs per 6 ml SM in running stock, they are unlikely to experience scramble competition; especially due to their reduced feeding rates [28]. Further, the marginal difference in the egg density used for the generation of standardized flies and that of running stocks is unlikely to influence our results. The egg collection from the selected and control populations were staggered by the developmental time difference so as to obtain similar aged adult flies for experimentation purpose [24].

At the time of starting these experiments, the control (JB) populations had passed through 232 generations of maintenance on 21 days, egg-to-egg discrete generation cycles, while the selected (FLJ) populations had been under simultaneous selection for faster pre-adult development and indirect selection for extended adult longevity for 126 generations.

## 2.3. Larva collection and fly media

Prior to the collection of synchronized eggs, standardized fly populations were supplemented with a generous amount of live-yeast and acetic acid paste for 3 days to boost their egg-laying. After 3 days, they were provided with a fresh sterile SM plate for 1 h and at the end of 1 h, the SM plate was replaced by an uncontaminated non-nutritive agar plate at every 1 h interval for three successive intervals. SM and first two agar plates were discarded and eggs laid on the third agar plate (fourth plate in the series), hereafter referred to as 'synchronized eggs' were used in all experiments unless otherwise mentioned (figure 1*c*).

The composition (table 1) and preparation of the standard media (SM) are as specified in Chandrashekara & Shakarad [31]. In the present experiments, in addition to SM for the maintenance of populations, liquid media (without agar-agar) was prepared to facilitate the sampling of larvae with ease and hence called liquid standard media (LSM) (table 1).

## 2.4. Critical size, post-critical duration and body size (in terms of weight)

Twenty-two hours post-egg-laying, 30 newly hatched (first instar) larvae were transferred to small Petri dishes (5.5 mm in diameter, Tarson) containing 2000 µl of LSM. Ten such plates were set up per population per time point and incubated at SLC. The same process was followed for other

experiments unless stated otherwise. Through pilot experiments, the average duration to attain the critical size (a.k.a. minimum viable weight/size for eclosion; MVW eclosion) was estimated to be 62 h and 74 h for the selected (FLJ) and control (JB) populations, respectively. Hence, the sampling of larvae for this experiment was initiated at 60 and 70 h for the selected and control populations, respectively. The sampling consisted of harvesting 300 larvae each from the selected and control populations. The larvae were washed with reverse osmosis (RO) water (in order to remove food particles sticking to the body surface) and rolled on a tissue towel to remove excess water. Thereafter, the larvae were sorted into 10 groups of 30 individuals each. Five groups were transferred to five pre-labelled vials containing 5 ml of non-nutritive agar and incubated at SLC, while the other five groups were weighed on Citizen (CM11) microbalance (figure 1c). The entire process of harvesting larvae, sorting them into batches, weighing and incubating was repeated at every 2 h interval till they started wandering. We adopted the criterion described in previous studies [10,11] with some modifications, for the calculation of critical size time point. The average critical size was the weight at which at least 50% of larvae undergo metamorphosis into adults even under non-availability of food. The total developmental time by which this weight is attained is referred to as 'critical duration'.

The average post-critical duration of both population types was estimated as the difference in the time lag between the attainment of critical size and the average duration to pupation under the availability of ad libitum food. Synchronized first instar larvae from non-nutritive agar plates were harvested, washed and transferred to 5 ml SM vials. The vials were incubated at SLC. At the pre-determined critical duration (estimated from previous experiment), the larvae were re-harvested and either transferred to non-nutritive agar vials or SM vials and incubated again under SLC. Five vials each with 30 larvae per treatment per population were set up. Time duration from critical duration till pupation was calculated by observing larvae at 2 h intervals till no further fresh pupations were observed. The number of pupae was scored and recorded. Emerging adults from both treatment vials were sorted based on gender and weighed on microbalance in order to estimate body size differences (in terms of weight).

## 2.5. Larval growth rate, food ingestion rate and development time

In this experiment, synchronized eggs were collected, transferred to Petri plates containing a thin film of LSM on 10% agar base and incubated under SLC. Triplicate sample of each of the six populations with 20 larvae per replicate per time point was washed, rolled on tissue towel and weighed on Citizen (CM11) microbalance at every 4 h interval till the pupation time point. The first reading was taken at 24 h from the mid-point of synchronized egg collection window—marked as zero-hour reading. The weight of an average single larva was deduced by dividing the group larval weight by the number of larvae.

Post-development of red eyespots in pupae, vials were checked at every 4 h interval for the emergence of adult flies. The emerging flies were collected into pre-labelled empty dry vials, sorted according to gender under mild $CO_2$ anaesthesia and recorded in data books. The mid-point between two successive 4 h checkpoints was taken as the time of emergence. The average development time was estimated from these primary data. The flies of a given treatment and gender were pooled and held in neatly labelled clean dry vials, freeze killed at −80°C, and five replicate groups of flies per gender per population were weighed to obtain the size of the fly measured as fresh weight.

A colourimetric assay was performed [32,33] with some modifications, to assess the food intake during different larval growth stages. For larval food ingestion assay, the newly hatched larvae from the synchronized batch of eggs were transferred to LSM in batches of 100 larvae per Petri dish. The Petri dishes were incubated at SLC in Power Scientific, USA, incubators. In all, there were nine such Petri dishes per population. Three Petri dishes per population were taken out after 12 h incubation, for L1 (mid-L1). Fifty larvae per population were harvested from the plates, washed with RO water, rolled over tissue towel and transferred to fresh agar plates overlaid with 2 ml double-distilled water. The larvae were allowed to be in double-distilled water for 10 min. After this starvation period of 10 min, the larvae were rolled on Kimwipe towel and transferred to fresh agar plate overlaid with 5 ml of 4% (w/v) blue dye (Erioglaucine disodium salt—Sigma Aldrich) mixed with 2 g yeast. The larvae were allowed an acclimatization period of 2 min, subsequent to which they were allowed to feed on the dye–yeast mix for 2 h. Immediately after 2 h interval, chilled (4°C) water was poured on to the larvae to arrest further feeding. Larvae were washed twice with distilled water to remove debris and rolled on Kimwipe tissue towel. A batch of 50 larvae was homogenized in 500 µl of PBS (1×). Samples were centrifuged at 135 000 r.p.m. (Eppendorf, 5430R) for 10 min. Then 100 µl of supernatant was used for optical density (OD) reading. Absorbance was taken at 625 nm on ELISA plate reader (ECIL micro scan MS5605A). The entire process from harvesting of larvae till measuring

of OD was repeated at 36 (mid-L2) and 48 h for selected, and 36 and 52 h for control populations post-transfer of freshly hatched L1 to LSM. The time durations chosen were believed to have caught the larvae in mid-L1, mid-L2 and L3 (prior to the attainment of the critical size) stages, respectively. The differential time point was chosen for the L3 so as to assess the larvae of similar physiological age [24].

## 2.6. Statistical analyses

Univariate analysis of variance, under general linear model (GLM) using SPSS v. 22 [34] was carried out on critical, post-critical and pupal duration; feeding rate and adult weight with treatment and selection regime as fixed factors, and replications as random factors as in Prasad *et al.* [24]. Since, in all cases, the population means were used as the units of analysis, only fixed-factor effects and interactions could be tested for significance [24].

To understand the impact of selection on the growth rate, linear regression analysis was performed on the larval stage-specific weight gain with L1–24 h: initial 6 data points, L2–24 h: data points 7–12 and L3: 13th data point and beyond [35]. The regression slope 'b' of the three larval stages were compared between selected and control populations using the *t*-test [36,37].

We also ascertained the impact of selection on body size distribution by fitting a normal probability density function, to capture the possible population distribution using actual data with the following equation:

$$f(x) = \frac{1}{\sigma 2\pi} e^{-}(x_i - \mu)/2\sigma^2,$$

where $\sigma$ represents standard deviation, $\sigma^2$ represents variance, $x$ represents mean ($\mu$), $x_i = (\mu + \sigma)$ or $(\mu + 2\sigma)$ or $(\mu + 3\sigma)$. $\pi = 3.14$, $e = 2.71$.

# 3. Results

## 3.1. Selection for accelerated development leads to the evolution of smaller critical size

There was a significant effect of selection on the critical size ($F_{1,2} = 24.45$, $p = 0.0385$; figure 2*a*) and critical duration ($F_{1,2} = 192.66$, $p = 0.0034$; figure 2*b*). The selected (FLJ) populations attained their critical size at an average wet weight of 1002.66 μg in an average duration of 62.5 h compared to their ancestral control (JB) whose average wet weight was 1308.71 μg attained in 74 h. A reduction of 23.38% in critical weight was attained with a reduction of 15.31% in critical developmental duration.

Further, there was a significant impact of selection on post-critical developmental duration ($F_{1,2} = 344.32$, $p = 0.003$; figure 2*c*). The developmental duration, post-attainment of critical size was reduced by 56.8% in the selected populations when compared with their ancestral control populations. Furthermore, there was no significant effect of the availability of food on the post-critical developmental duration of larvae in both the selected and control populations ($F_{1,2} = 0.763$, $p = 0.473$; figure 2*c*). In addition, the reduction in the pupal duration was also non-significant ($F_{1,2} = 5.960$, $p = 0.135$; figure 2*d*) between the selected (86.36 h) and control (89.96 h) populations. Overall, the egg to adult development time significantly ($F_{1,2} = 363.701$, $p = 0.003$; figure 2*d*) reduced by 17.5% in selected populations compared to their ancestral controls. An average adult from populations under selection for faster pre-adult development took 188.34 h to eclose from the egg, while control populations took 228.3 h to eclose.

## 3.2. Selection for accelerated pre-adult development affects larval growth rate at second instar

The reduction in the critical size associated with a reduction in larval developmental duration of the selected compared to the control populations could be a correlated response without any change in the larval growth rate. To address this, the larval growth (measured as wet weight) trajectories of the two population types were ascertained at every 4 h interval from the time of hatching till pupation (figure 3*a*). Linear regression analysis of the three larval stages showed no significant difference in the slope during the L1 (first 24 h, $t = 0.98$; figure 3*a*) and L3 stages (post 48 h till wandering stage mid-point, $t = 0.16$; figure 3*a* and table 2). However, the slope of the selected populations was significantly higher than that of their ancestral control during the L2 stage ($t = 3.54$, $p < 0.01$; figure 3*a* and table 2). The increased growth rate was not due to an increase in food intake that was not significantly different between the selected and control populations ($F_{1,2} = 16.14$, $p = 0.057$; figure 3*b*).

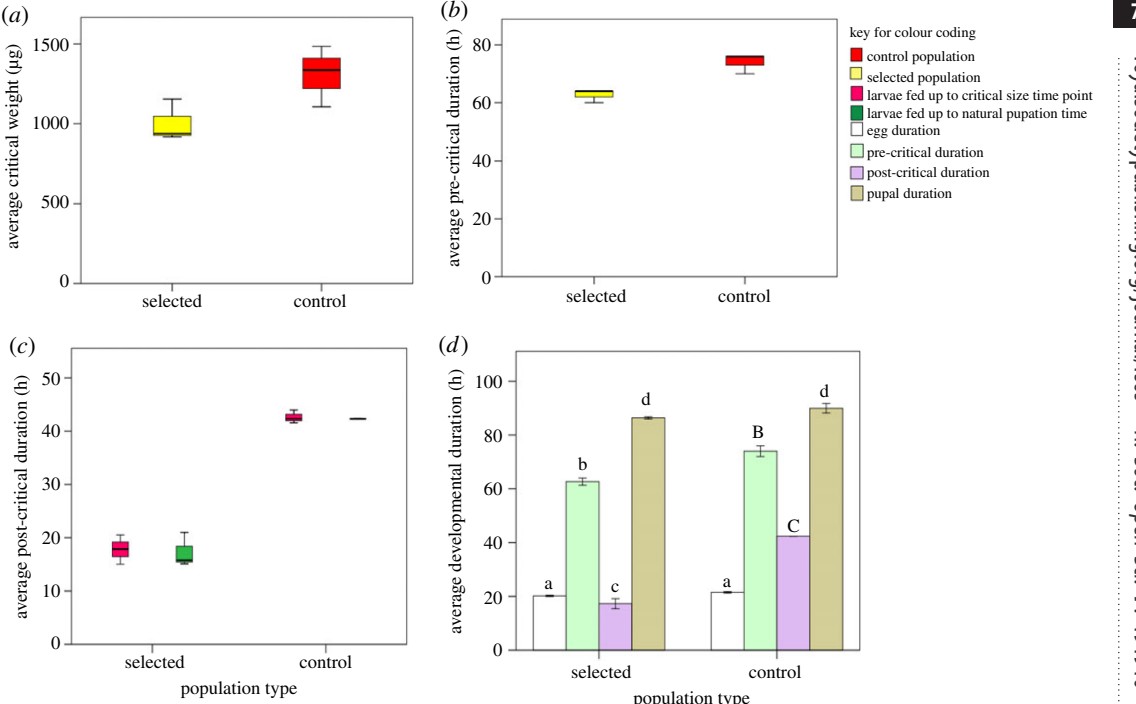

**Figure 2.** Average (±s.e) of larval (a) critical weight (representative of critical size), (b) pre-critical duration, (c) post-critical duration under availability and non-availability of food post-attainment of critical size and (d) developmental duration from egg to adult eclosion. Stage-specific comparisons are made between selected and control populations. Bars with small and capital case alphabets represent significant differences, while those with small case same alphabets are not significantly different.

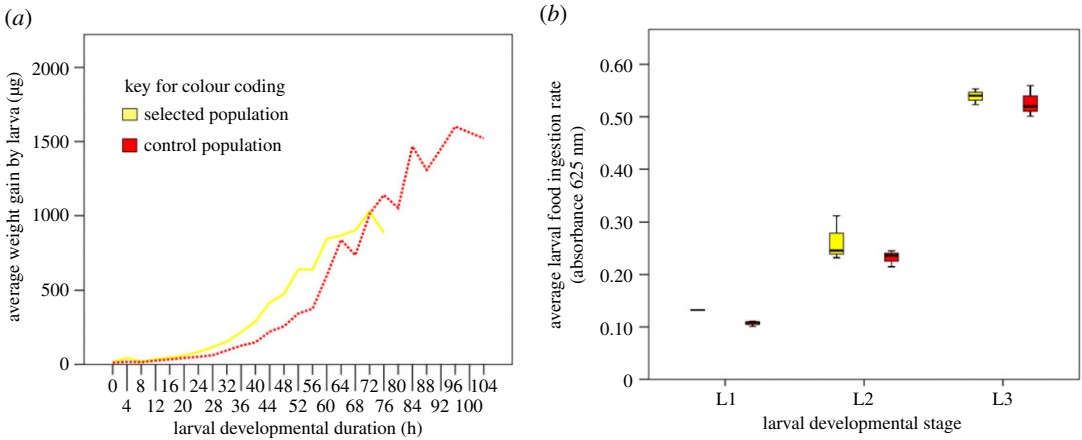

**Figure 3.** Average (±s.e) (a) larval growth rate in terms of weight gain and (b) larval food ingestion rate during mid-L1, L2 and L3 stages.

## 3.3. Impact of selection for accelerated pre-adult development on adult body size and its distribution

We found a significant reduction ($F_{1,2} = 35.682$, $p = 0.027$; figure 4a) in the fresh/wet weight of adults due to selection for faster pre-adult development. There was a reduction of 20.14% in the wet weight of an average fly from the selected populations (921.23 µg) in comparison to an average fly from control population (1153.64 µg) when they had access to ad libitum food. Further, there was a significant effect of feeding regimen on the wet weight of the flies ($F_{1,2} = 498.54$, $p = 0.002$; figure 4a). The overall reduction in the weight of the flies that emerged after feeding only up to critical duration in

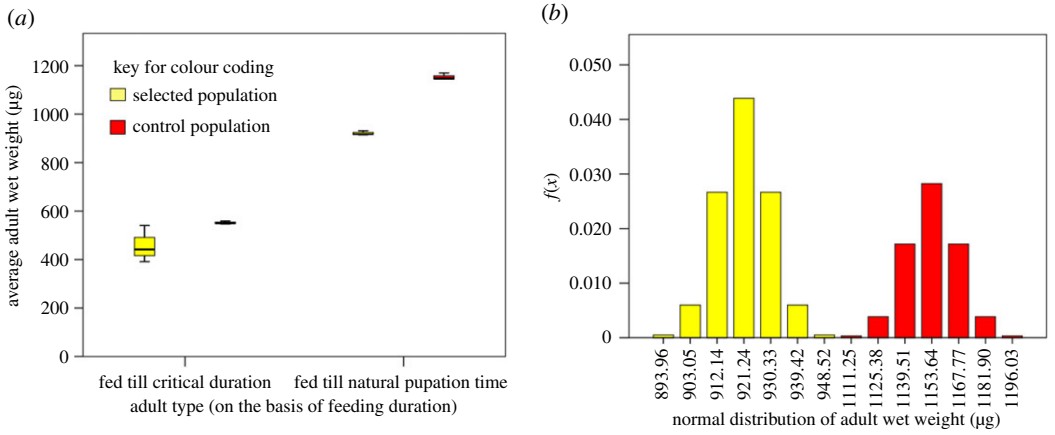

**Figure 4.** (*a*) Average (±s.e) adult wet weight (representative of adult body size) and effect of food availability up to critical size and natural pupation time in selected and control populations and (*b*) normal probability density distribution of adult body weight.

**Table 2.** Regression table: $\beta$-value of control and selected populations and respective *t*-values at different larval stages.

| larval duration—$\beta$-values | control populations | selected populations | *t*-values |
|---|---|---|---|
| L1—zero to 24 h post-hatching | 1.66 | 2.17 | 0.97 |
| L2—24–48 h post-hatching | 8.62 | 17.21 | 3.54 |
| L3—48 h to wandering stage mid-point[a] | 49.77 | 29.49 | 0.16 |

[a]For selected populations, as L3 is of small duration, thus mid-point value is considered for the analysis. L1, L2 and L3 stand for first, second and third larval stages.

comparison to those that fed till they naturally wandered off to pupate was 51.33%. There was no selection × feeding duration interaction effect ($F_{1,2} = 0.031$, $p = 0.877$).

There was no significant difference in the weight of the adults that emerged from larvae fed up to critical time (figure 4*a*). In *D. melanogaster*, adult body size is tightly correlated with development time. Since there was a reduction in total development duration of selected populations by 17.5%, we were interested in ascertaining the degree of shift in the adult body distributions. Hence, we constructed a distribution using a normal probability density function. The normal probability distribution of the adult size (measured as wet weight at emergence) of the flies from the selected populations was shifted to the left of the normal probability distribution of control population flies (figure 4*b*) and showed no overlap.

## 4. Discussion

In *D. melanogaster*, although intricate regulatory hierarchy is suggested to respond to variations in nutrient availability and ensure uniformity of species-specific final body size [38], studies selecting for faster pre-adult development have reported a significant reduction in body size [22,24,39]. The reduced adult size concomitantly resulted in reduced fecundity [23]. However, in an earlier study, we reported higher fecundity in flies that were significantly smaller than their ancestral controls [40]. In *D. melanogaster*, it has also been reported that much of the resources that are used for survival during metamorphosis and early adult activities are acquired in the late L3 stage, post-attainment of critical size and are stored in larval fat bodies [1,2]. However, the post-critical duration in the populations selected for faster pre-adult development has significantly reduced, suggesting that these populations are likely to have evolved mechanisms such as energy acquisition, storage and utilization during the adult stage due to long adult life owing to the selection protocol, and thus have fitness comparable to their ancestral control populations [40]. Incidentally, there was no fitness cost in terms of viability during the pre-adult stages (electronic supplementary material, figure S1), suggesting that the long adult lifespan seems to have mitigated the viability cost during the pre-adult stage of our selected

faster developing, late reproducing and JB-derived (FLJ) populations, unlike that reported by Prasad *et al*. [24] in their FEJ populations selected for faster pre-adult development and early reproduction.

Adult body size in insects is an important fitness governing trait that is determined by three factors: (i) the number of larval instars, (ii) the size increment at each larval moult, and (iii) the size at which the last larval instar stops feeding and initiates metamorphosis (a.k.a. critical size) [41]. An additional factor that can alter the final adult size is the duration of each of the larval stages. A reduction in the duration of the first and third larval instar contributing to the reduction in final adult size has been reported by Prasad *et al*. [24]. Further, Prasad *et al*. [24] suggested that the critical size in *D. melanogaster* can evolve under selection for faster pre-adult development, but they did not have direct evidence of this. Here, we show that populations under selection for faster pre-adult development and late reproduction (FLJs) have evolved significantly smaller critical size (figure 2*a*) along with significant reduction in developmental duration supporting the Prasad *et al*. [24] hypothesis that direct selection for faster pre-adult development results in smaller adult body size via reduction in development duration [23] and smaller critical size. Hironaka *et al*. [18] using nine *Drosophila* species showed that variation in organism size can solely be explained by species-specific critical size. *Drosophila melanogaster* inhabits ephemeral environment where nutritional conditions are deteriorating continuously and thus is under constant pressure of faster development. Larvae exposed to poor dietary condition since the start of larval life exhibit higher metabolic efficacy and accelerated development rate through evolution of smaller critical size [25]. The yellow dung fly *Scathophaga stercoraria*, like *D. melanogaster*, adopts adaptive bailout under the condition of continuously deteriorating environment and accelerates its development along with lower critical size attainment indicating critical weight to be the target of selection in *S. stercoraria* [42].

As opposed to a previous study where lines selected for small body size grew slowly [26], populations under selection for faster pre-adult development in this study demonstrated higher growth rate than control populations during the second larval instar. There was no significant difference in the larval growth rate of the selected populations and their ancestral control populations during the first as well as the third larval instars (figure 3*b*). The higher growth rate could possibly be another reason for the lack of pre-adult viability cost in our selected populations (electronic supplementary material, figure S1). The higher pre-adult mortality in Prasad *et al*. [24] could perhaps be due to lower growth rate in their selected (FEJ) populations. Further, a reduction of 23.38% in critical weight with a reduction of only 15.31% in critical developmental duration suggests that the control (JB) populations are gaining disproportionately higher weight during the last 12 h prior to reaching critical size (electronic supplementary material, table S1). Unlike the study of Prasad *et al*., where selection for faster development leads to a reduction in pupal duration [24], we found no significant difference in the pupal duration of our selected (FLJ) populations (figure 2*d*). The differences in the results of the two studies despite the ancestral populations being the same could be due to the differences in the selection pressure in the adult phase. The FEJ populations [24] are under pressure to maximize their fitness by day 3 post-emergence while our selected (FLJ) populations have long adult life. The importance of such cross-life-stage effects in life-history evolution have been reviewed in Prasad & Joshi [43].

The adult body size in *D. melanogaster* is a highly plastic trait, influenced by both genotype and the environment that helps the organisms to survive fluctuating food availability, both quantitatively and qualitatively [10,19]. Since the *Drosophila* adult is post-mitotic, the size is primarily determined by the larval duration and larval feeding rate that would indicate the amount of food ingested. The amount of food ingested during mid-L1, L2 and L3 stages were not significantly different in our selected (FLJ) populations when compared with their ancestral control (JB) populations (figure 3*b*). Two independent studies had shown reduced larval feeding rates as a correlated response to selection for faster pre-adult development [24,28]. Rajamani *et al*. [28] had reported a reduction in the feeding rate of the selected (FLJ) populations compared to the control (JB) populations during relatively early stages (10 and 32 generations) of selection. The differences in the results obtained in the present study and those of Prasad *et al*. [24] and Rajamani *et al*. [28] could be due to different aspects of feeding efficiency being measured and/or that our selected (FLJ) populations have regained their feeding rates during subsequent selection cycles due to the relaxed selection during the adult life.

Vijendravarma *et al*. [25] also reported no difference in the growth trajectories of their selected (for chronic malnutrition) and control populations till attainment of critical size. However, the growth rate of the faster developing populations in our study was significantly higher during the second larval instar, unlike that of Partridge *et al*. [26] study where the growth rate was lower for the smaller adult size selections and higher for the larger adult size selections. The increased growth rate during second

larval instar accompanied by reduced critical size could be responsible for the significant reduction in the post-critical duration of the larvae in our selected populations. The observed differences in the growth rates in the three studies might be due to the difference in the genotypes of the populations under study and/or genuinely indicate the dynamic nature of responses due to different component traits being the target of selection. The reduction in the adult body size could be due to the reduction in the critical size and/or reduction in the post-critical duration. This could be another form of adaptive bailout as in the case of *S. stercoraria* in response to food limitation [42]. The populations under selection for faster development might be exhibiting adaptive bailout [42] not due to external food limitation but due to an internal trigger.

# 5. Conclusion

Overall, our study provides empirical evidence for reduction in critical size underlying final adult body size reduction in populations of *D. melanogaster* under selection for faster pre-adult development. Additionally, we provide experimental evidence for cross-life-stage effects in life-history evolution in *D. melanogaster*.

Data accessibility. Our entire data (both main data and electronic supplementary material, supporting data) are deposited at the Dryad Digital Repository: https://doi.org/10.5061/dryad.k6djh9w32 [44]. The average (±s.e.) pre-adult survival and average (±s.e.) weight gain during last 2 h prior to critical size, in selected and control populations, are uploaded as electronic supplementary material.

Authors' contributions. M.N.S. and K.S. conceived and designed the study. K.S. and N.M. collected data and performed the experiments. K.S. and M.N.S. analysed the data and wrote the manuscript. All authors gave final approval for publication.

Competing interests. The authors declare no competing interests.

Funding. This work was supported by Council for Scientific and Industrial Research (CSIR-grant no. 37(1495/11/EMR-II)) for doctoral funding and University of Delhi (R&D grant no. DU: DURC/2014–2015).

Acknowledgements. We would like to thank the editor, an anonymous reviewer and Prof. Amitabh Joshi, JNCASR, Bengaluru, for their valuable comments on earlier versions of manuscript and funding agencies for their financial support.

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
