## [Reviewer comments · Royal Society Open Science]

Review History

RSOS-191910.R0 (Original submission)

Review form: Reviewer 1

Is the manuscript scientifically sound in its present form?

No

Are the interpretations and conclusions justified by the results?

Yes

Is the language acceptable?

No

Do you have any ethical concerns with this paper?

No

Have you any concerns about statistical analyses in this paper?

Yes

Recommendation?

Major revision is needed (please make suggestions in comments)

Comments to the Author(s)

This is an interesting paper that explores the effect of prolonged selection for accelerated developmental-time on the critical size phenomenon, using *Drosophila melanogaster* as a model organism. The authors selected for accelerated developmental time and extended longevity across 231 generations, resulting in a reduction in adult body size and egg-to-adult developmental time. This is in turn correlated with a reduction in critical size, and pre- and post-critical size developmental time, as well as second-instar growth rates. The results of the research are generally clear and support the hypothesis that selection for accelerated development reduces critical size.

Nevertheless, there are a number of relatively substantive problems with the MS that need to be addressed before publication.

1) The authors suggest that the terms minimal viable weight (MVW) and critical size (CS) are interchangeable, while several researchers have demonstrated that they are distinct phenomena. The authors correctly define CS as the size above which starvation does not alter the time course to metamorphosis. The CS is therefore a physiological switch in the effect of nutrition on the hormone cascade that ends on metamorphosis, and the molecular genetic mechanisms underlying the switch have been well elucidated. The MVW for pupariation (MVW_{pupariation}) is the minimum weight at which >50% of starved larvae pupariate, while the MVW for eclosion (MVW_{eclosion}) is the minimum weight at which >50% of starved larvae eclose as adults. Both of these therefore reflect a larva's nutritional status and its ability to mobilize stored resources and survive to different points in development upon starvation. While the three phenomena are correlated, they are not the same. Generally MVW_{pupariation} < MVW_{eclosion} < CS (see Stieper et al 2008 and Hironaka et al 2019 for more on this). Further, only *Drosophila* researchers confound CS and MVW: in *M. sexta* for example, CS is the time at which starvation no longer affects the time course to metamorphosis.

While I think that it is fine for the authors to use MVW_{eclosion} as a proxy for CS (as it has been used in various publications, e.g. Mirth et al 2014, Hironaka et al 2019), it adds confusion if the distinction is not clearly stated in the text. Further, it is conceivable that an observed reduction in MVW_{eclosion} may not be accompanied by a reduction in CS *sensu stricto*. Problematically, while the CS phenomenon is well understood at a physiological level, the MVW phenomenon is not, although it likely involves the initiation of ecdysteroidgenesis by factors other than nutrition, through a bail-out response or a 'leaky' prothoracic gland (see Nijhout et al 2014 WIRE Dev Bio for more details). Thus, while the observed response to selection may indeed reflect changes in the mechanisms that initiate ecdysteroidgenesis at attainment of CS, it is also possible that they reflect changes in the ability to initiate a bail-out response, although these mechanisms remain unknown. Without distinguishing between CS and MVW, the distinction is lost.

2) L64 "larvae monitor their size by the relative growth of various organs in proportion to each other [4]". The cited paper does not support the statement (indeed I do not know of any paper that does).

3) L67 "Critical size is suggested to evolve in response to environmental conditions. If the environmental conditions are conducive, then slower growth along with larger body size as a consequence of larger critical size is favoured by natural selection; while under non-conducive environment, faster growth with the smaller critical size is selected". This statement has unclear logic. All traits evolve by natural selection in response to 'environmental conditions [20]'. The question is what is the selective pressure and what is it targeting. Does low nutrition select for smaller body size or accelerated growth? And does it target critical size directly, or is critical size a consequence of selection on body size/growth rate? See Hironaka et al 2019 for more details on

how selection may act to change critical size. Further, the citation is for amphibians and not holometabolous insects.

4) L80. Please provide citations for the selection experiments.

5) L90. Here you correctly argue that the change in critical size is a correlated response to selection for accelerated development. However, in the title you state "Higher instantaneous growth during second larval instar leads to lower critical size in *Drosophila melanogaster* populations", which is not supported by your data, since both growth rate and change in critical size are a correlated response to changes in developmental time. The title should be changed to reflect this.

6) In the methods you describe the selection regime, and provide a helpful figure. However, you do not explicitly state how many generations you have maintained selection apart from in the introduction. You should state this in the M&M section also.

7) L130. This section is a little difficult to follow. Problematically you use the term 'standardized flies' (L138) before you define it (L143).

8) Minor point: The sampling method for calculating MVW/CS is rather complex. A figure would be helpful to show how larvae were sampled at each time point.

9) L249. The authors report the MVW/CS as 1002.66 μ g which is likely much more precise than the microbalance that they are using. It would be useful to adjust the number of significant figures to reflect the precision of the microbalance.

10) L256 "Furthermore, there was no significant effect of the availability of food on the post-critical developmental duration of larvae in both the selected and control populations" This is an interesting result since many studies have shown that starvation at CS accelerates pupariation in *Drosophila* (e.g. Steiper et al 2009). Does this reflect the relatively imprecise method used to calculate CS/MVW? The challenge with the methodology used is that even though eggs were collected in a 1h cohort, developmental stage can vary considerably by the time larvae get to the 3rd instar (Ashburner discusses this in his book). Thus the average mass of a cohort of larvae at 70h AEL may hide considerable variation in size. Further, the authors only calculated MVW/CS at a resolution of every 2 hours. Other methods to calculate MVW/CS (e.g. using logistic regression, Steiper et al 2009, Hironaka et al 2019) appear to be more precise and may have been able to detect the acceleration in developmental time upon starvation at CS/MVW.

11) L305 "However, the post-critical duration in the populations selected for faster pre-adult development has significantly reduced- suggesting that these populations are likely to have evolved mechanisms so as to have higher fitness compared to their ancestral control populations." I do not follow the logic of this argument. Why would this lead to higher fitness? Does the author mean higher fitness under the specified selection pressure?

12) L312 "A couple of studies in *Drosophila* [10, 11] have shown the critical size to be static with respect to nutrition, thus supporting the above suggestion. However, many studies over the past three decades have shown a reduction in the adult size of *D. melanogaster* flies in response to lack of nutrition [25], increased growth temperatures [12] and selection for faster pre-adult development [24, 39]." This is rather misleading since the authors confound phenotypic plasticity with adaptation. The cited studies [1-,11] suggest that CS/MVW does not vary within a genotype in response to nutrition; that is, it is not nutritionally plastic. This does not mean that there is no genetic variation for CS/MVW nor that it cannot evolve.

13) L323 "Here we show that populations under selection for faster pre-adult development have evolved significantly smaller critical size (Figure 2A) supporting the view that critical size in *D. melanogaster* is polyphenic under different growth conditions [19, 25]". Again, the authors

confound plasticity with adaptation. Polyphenism is the response of a phenotype to environmental conditions (strictly speaking two or more discrete phenotypes, e.g. castes of bees). Their data support the hypothesis that CS/MVW is genetically variable and can respond to selection, not that it is phenotypically plastic.

14) I encourage the authors to revise their figures. The colors are rather difficult to follow (a key in the figure would help). Also, it is increasingly standard to include all the data points in box plots and bar charts, to give the reader an idea of the distribution of data. Finally, the authors must specify what the error bars are and provide sample sizes for all statistical tests.

15) Statistical Analysis. Details of the statistical analyses need to be included. What statistical models were used to analyze the data? The authors used a GLM but did not specify the link function or probability distribution. How were the regression slopes compared? Why fit a linear regression when you state clearly in the introduction that growth prior to CS/MVW is exponential? Did you test for homoscedasticity?

16) L289 "In *D. melanogaster* body size is tightly correlated with development time, thus one would expect the body size distributions of the selected and control populations to be non-overlapping". Why would you expect it to be non-overlapping? Different, possible, but not necessarily non-overlapping. Also, I am not sure why the authors examined the impact of selection on body size distribution. Why not just apply a standard two-sample t-test to see whether body size is different?

Review form: Reviewer 2 (Amitabh Joshi)

Is the manuscript scientifically sound in its present form?

Yes

Are the interpretations and conclusions justified by the results?

No

Is the language acceptable?

Yes

Do you have any ethical concerns with this paper?

No

Have you any concerns about statistical analyses in this paper?

No

Recommendation?

Accept with minor revision (please list in comments)

Comments to the Author(s)

This is a nice piece of work. The experimental design is good and statistics are appropriate. The issue is one of interest, especially within the fly experimental evolution community. My main concerns are with the way the Introduction and Discussion are framed, which should be improved in a revision. I am attaching an annotated copy of the manuscript (Appendix A) with detailed comments to this end. Since I choose not to be anonymous, I urge the authors to feel free to contact me if they have any doubts when they undertake a revision.

Decision letter (RSOS-191910.R0)

27-Feb-2020

Dear Dr SHAKARAD,

The editors assigned to your paper ("Higher instantaneous growth during second larval instar leads to lower critical size in *Drosophila melanogaster* populations") have now received comments from reviewers. We would like you to revise your paper in accordance with the referee and Associate Editor suggestions which can be found below (not including confidential reports to the Editor). Please note this decision does not guarantee eventual acceptance.

Please submit a copy of your revised paper before 21-Mar-2020. Please note that the revision deadline will expire at 00.00am on this date. If we do not hear from you within this time then it will be assumed that the paper has been withdrawn. In exceptional circumstances, extensions may be possible if agreed with the Editorial Office in advance. We do not allow multiple rounds of revision so we urge you to make every effort to fully address all of the comments at this stage. If deemed necessary by the Editors, your manuscript will be sent back to one or more of the original reviewers for assessment. If the original reviewers are not available, we may invite new reviewers.

- Data accessibility

<http://datadryad.org/submit?journalID=RSOS&manu=RSOS-191910>

- **Competing interests**

- **Authors' contributions**

- **Acknowledgements**

- **Funding statement**

on behalf of Professor Laura Johnston (Associate Editor) and Kevin Padian (Subject Editor)
openscience@royalsociety.org

Associate Editor's comments (Professor Laura Johnston):

Associate Editor: 1

Comments to the Author:

Two expert referees have now reviewed your work, "Higher instantaneous growth during second larval instar leads to lower critical size in *Drosophila melanogaster* populations". Both reviewers felt the work is of high quality and of significant interest, however, several changes are recommended prior to publication. Review 1, in particular, notes several important points that need to be addressed. Reviewer 2 suggests making revisions to the text and has included suggestions in the text itself, including more details regarding statistical methods.

Reviewers' Comments to Author:

Reviewer: 1

Comments to the Author(s)

This is an interesting paper that explores the effect of prolonged selection for accelerated developmental-time on the critical size phenomenon, using *Drosophila melanogaster* as a model organism. The authors selected for accelerated developmental time and extended longevity across 231 generations, resulting in a reduction in adult body size and egg-to-adult developmental time. This is in turn correlated with a reduction in critical size, and pre- and post-critical size developmental time, as well as second-instar growth rates. The results of the research are generally clear and support the hypothesis that selection for accelerated development reduces critical size.

Nevertheless, there are a number of relatively substantive problems with the MS that need to be addressed before publication.

1) The authors suggest that the terms minimal viable weight (MVW) and critical size (CS) are interchangeable, while several researchers have demonstrated that they are distinct phenomena. The authors correctly define CS as the size above which starvation does not alter the time course to metamorphosis. The CS is therefore a physiological switch in the effect of nutrition on the hormone cascade that ends on metamorphosis, and the molecular genetic mechanisms underlying the switch have been well elucidated. The MVW for pupariation (MVW_{pupariation}) is the minimum weight at which >50% of starved larvae pupariate, while the MVW for eclosion (MVW_{eclosion}) is the minimum weight at which >50% of starved larvae eclose as adults. Both of these therefore reflect a larva's nutritional status and its ability to mobilize stored resources and survive to different points in development upon starvation. While the three phenomena are correlated, they are not the same. Generally MVW_{pupariation} < MVW_{eclosion} < CS (see Stieper et al 2008 and Hironaka et al 2019 for more on this). Further, only *Drosophila* researchers confound CS and MVW: in *M. sexta* for example, CS is the time at which starvation no longer affects the time course to metamorphosis.

While I think that it is fine for the authors to use MVW_{eclosion} as a proxy for CS (as it has been used in various publications, e.g. Mirth et al 2014, Hironaka et al 2019), it adds confusion if the distinction is not clearly stated in the text. Further, it is conceivable that an observed reduction in MVW_{eclosion} may not be accompanied by a reduction in CS *sensu stricto*. Problematically, while the CS phenomenon is well understood at a physiological level, the MVW phenomenon is not, although it likely involves the initiation of ecdysteroidgenesis by factors other than nutrition, through a bail-out response or a 'leaky' prothoracic gland (see Nijhout et al 2014 WIRE Dev Bio for more details). Thus, while the observed response to selection may indeed reflect changes in the mechanisms that initiate ecdysteroidgenesis at attainment of CS, it is also possible that they reflect changes in the ability to initiate a bail-out response, although these mechanisms remain unknown. Without distinguishing between CS and MVW, the distinction is lost.

2) L64 "larvae monitor their size by the relative growth of various organs in proportion to each other [4]". The cited paper does not support the statement (indeed I do not know of any paper that does).

3) L67 "Critical size is suggested to evolve in response to environmental conditions. If the environmental conditions are conducive, then slower growth along with larger body size as a consequence of larger critical size is favoured by natural selection; while under non-conducive environment, faster growth with the smaller critical size is selected". This statement has unclear logic. All traits evolve by natural selection in response to 'environmental conditions [20]'. The question is what is the selective pressure and what is it targeting. Does low nutrition select for smaller body size or accelerated growth? And does it target critical size directly, or is critical size a consequence of selection on body size/growth rate? See Hironaka et al 2019 for more details on

how selection may act to change critical size. Further, the citation is for amphibians and not holometabolous insects.

4) L80. Please provide citations for the selection experiments.

5) L90. Here you correctly argue that the change in critical size is a correlated response to selection for accelerated development. However, in the title you state "Higher instantaneous growth during second larval instar leads to lower critical size in *Drosophila melanogaster* populations", which is not supported by your data, since both growth rate and change in critical size are a correlated response to changes in developmental time. The title should be changed to reflect this.

6) In the methods you describe the selection regime, and provide a helpful figure. However, you do not explicitly state how many generations you have maintained selection apart from in the introduction. You should state this in the M&M section also.

7) L130. This section is a little difficult to follow. Problematically you use the term 'standardized flies' (L138) before you define it (L143).

8) Minor point: The sampling method for calculating MVW/CS is rather complex. A figure would be helpful to show how larvae were sampled at each time point.

9) L249. The authors report the MVW/CS as 1002.66 μ g which is likely much more precise than the microbalance that they are using. It would be useful to adjust the number of significant figures to reflect the precision of the microbalance.

10) L256 "Furthermore, there was no significant effect of the availability of food on the post-critical developmental duration of larvae in both the selected and control populations" This is an interesting result since many studies have shown that starvation at CS accelerates pupariation in *Drosophila* (e.g. Steiper et al 2009). Does this reflect the relatively imprecise method used to calculate CS/MVW? The challenge with the methodology used is that even though eggs were collected in a 1h cohort, developmental stage can vary considerably by the time larvae get to the 3rd instar (Ashburner discusses this in his book). Thus the average mass of a cohort of larvae at 70h AEL may hide considerable variation in size. Further, the authors only calculated MVW/CS at a resolution of every 2 hours. Other methods to calculate MVW/CS (e.g. using logistic regression, Steiper et al 2009, Hironaka et al 2019) appear to be more precise and may have been able to detect the acceleration in developmental time upon starvation at CS/MVW.

11) L305 "However, the post-critical duration in the populations selected for faster pre-adult development has significantly reduced- suggesting that these populations are likely to have evolved mechanisms so as to have higher fitness compared to their ancestral control populations." I do not follow the logic of this argument. Why would this lead to higher fitness? Does the author mean higher fitness under the specified selection pressure?

12) L312 "A couple of studies in *Drosophila* [10, 11] have shown the critical size to be static with respect to nutrition, thus supporting the above suggestion. However, many studies over the past three decades have shown a reduction in the adult size of *D. melanogaster* flies in response to lack of nutrition [25], increased growth temperatures [12] and selection for faster pre-adult development [24, 39]." This is rather misleading since the authors confound phenotypic plasticity with adaptation. The cited studies [1-,11] suggest that CS/MVW does not vary within a genotype in response to nutrition; that is, it is not nutritionally plastic. This does not mean that there is no genetic variation for CS/MVW nor that it cannot evolve.

13) L323 "Here we show that populations under selection for faster pre-adult development have evolved significantly smaller critical size (Figure 2A) supporting the view that critical size in *D. melanogaster* is polyphenic under different growth conditions [19, 25]". Again, the authors

confound plasticity with adaptation. Polyphenism is the response of a phenotype to environmental conditions (strictly speaking two or more discrete phenotypes, e.g. castes of bees). Their data support the hypothesis that CS/MVW is genetically variable and can respond to selection, not that it is phenotypically plastic.

14) I encourage the authors to revise their figures. The colors are rather difficult to follow (a key in the figure would help). Also, it is increasingly standard to include all the data points in box plots and bar charts, to give the reader an idea of the distribution of data. Finally, the authors must specify what the error bars are and provide sample sizes for all statistical tests.

15) Statistical Analysis. Details of the statistical analyses need to be included. What statistical models were used to analyze the data? The authors used a GLM but did not specify the link function or probability distribution. How were the regression slopes compared? Why fit a linear regression when you state clearly in the introduction that growth prior to CS/MVW is exponential? Did you test for homoscedasticity?

16) L289 "In *D. melanogaster* body size is tightly correlated with development time, thus one would expect the body size distributions of the selected and control populations to be non-overlapping". Why would you expect it to be non-overlapping? Different, possible, but not necessarily non-overlapping. Also, I am not sure why the authors examined the impact of selection on body size distribution. Why not just apply a standard two-sample t-test to see whether body size is different?

Reviewer: 2

Comments to the Author(s)

This is a nice piece of work. The experimental design is good and statistics are appropriate. The issue is one of interest, especially within the fly experimental evolution community. My main concerns are with the way the Introduction and Discussion are framed, which should be improved in a revision. I am attaching an annotated copy of the manuscript with detailed comments to this end. Since I choose not to be anonymous, I urge the authors to feel free to contact me if they have any doubts when they undertake a revision.

Author's Response to Decision Letter for (RSOS-191910.R0)

See Appendix B.

RSOS-191910.R1 (Revision)

Review form: Reviewer 1

Is the manuscript scientifically sound in its present form?

Yes

Are the interpretations and conclusions justified by the results?

Yes

Is the language acceptable?

Yes

Do you have any ethical concerns with this paper?

No

Have you any concerns about statistical analyses in this paper?

No

Recommendation?

Accept with minor revision (please list in comments)

Comments to the Author(s)

1) The authors state: "In holometabolous and some hemimetabolous insects, the process of initiation of metamorphosis is dependent on attaining a certain minimum threshold size called critical size [7-10] beyond which starvation does not alter the time course to metamorphosis [10, 11-17]. Critical size, also called minimum critical size, is similar to minimal viable weight for eclosion in *Drosophila melanogaster* [16, 18]. Hence, throughout this study we have used the term 'critical size/weight'."

While I appreciate the authors clarifying the difference between MVW[eclosion] and critical size, this change rather misses the point. The authors define the critical size as the point at which starvation does not alter the time course to metamorphosis, but do not measure this. Rather they measure the size at which 50% of starved larvae survive to eclosion, which is the MVW[eclosion]. It is imprecise to say that MVW[eclosion] is 'similar' to critical size/weight. Rather *Drosophilists* use MVW[eclosion] as a proxy for critical size weight, which is what the authors do in their study. Thus the penultimate sentence of this paragraph should read something like: "In *Drosophila*, the size at which 50% of starved larvae successfully eclose as adults (the minimal viable weight for eclosion) is used as a proxy for critical size [16, 18]. We use this proxy for critical size/weight in this study."

2) The authors state: '...in *Manduca sexta* - a Lepidopteran holometabolous insect, larvae between 4th instar and 5th instar stage, whose head capsule size was greater than 5.4 mm were able to successfully pupate else undergo one more molt to 6th instar thus monitor their size by the growth of head capsule in proportion to body size.'

This is a slight overstatement of the conclusions of the cited paper. Nijhout observed a correlation between head capsule size and probability of pupating, and hypothesized that the larvae are monitoring their size by growth of the head capsule (which only grows between instars). However, this was a hypothesis. I am aware of no subsequent study that has explored this hypothesis further. I would therefore modify this sentence to say that the data 'suggest' that larvae monitor their size by the growth of head capsule.

Decision letter (RSOS-191910.R1)

12-May-2020

Dear Dr SHAKARAD:

On behalf of the Editors, I am pleased to inform you that your Manuscript RSOS-191910.R1 entitled "Evolution of reduced minimum critical size as a response to selection for rapid pre-adult development in *Drosophila melanogaster*." has been accepted for publication in Royal Society Open Science subject to minor revision in accordance with the referee suggestions. Please find the referees' comments at the end of this email.

The reviewers and Subject Editor have recommended publication, but also suggest some minor revisions to your manuscript. Therefore, I invite you to respond to the comments and revise your manuscript.

- Ethics statement

- Data accessibility

If you wish to submit your supporting data or code to Dryad (<http://datadryad.org/>), or modify your current submission to dryad, please use the following link:
<http://datadryad.org/submit?journalID=RSOS&manu=RSOS-191910.R1>

- Competing interests

- Authors' contributions

- Acknowledgements

- Funding statement

Please note that we cannot publish your manuscript without these end statements included. We have included a screenshot example of the end statements for reference. If you feel that a given

heading is not relevant to your paper, please nevertheless include the heading and explicitly state that it is not relevant to your work.

Because the schedule for publication is very tight, it is a condition of publication that you submit the revised version of your manuscript before 21-May-2020. Please note that the revision deadline will expire at 00.00am on this date. If you do not think you will be able to meet this date please let me know immediately.

on behalf of Professor Laura Johnston (Associate Editor) and Kevin Padian (Subject Editor)
openscience@royalsociety.org

Reviewer comments to Author:

Reviewer: 1

Comments to the Author(s)

1) The authors state: "In holometabolous and some hemimetabolous insects, the process of initiation of metamorphosis is dependent on attaining a certain minimum threshold size called critical size [7-10] beyond which starvation does not alter the time course to metamorphosis [10, 11-17]. Critical size, also called minimum critical size, is similar to minimal viable weight for eclosion in *Drosophila melanogaster* [16, 18]. Hence, throughout this study we have used the term 'critical size/weight'."

While I appreciate the authors clarifying the difference between MVW[eclosion] and critical size, this change rather misses the point. The authors define the critical size as the point at which starvation does not alter the time course to metamorphosis, but do not measure this. Rather they measure the size at which 50% of starved larvae survive to eclosion, which is the MVW[eclosion]. It is imprecise to say that MVW[eclosion] is 'similar' to critical size/weight. Rather *Drosophilists* use MVW[eclosion] as a proxy for critical size weight, which is what the authors do in their study. Thus the penultimate sentence of this paragraph should read something like: "In *Drosophila*, the size at which 50% of starved larvae successfully eclose as adults (the minimal viable weight for eclosion) is used as a proxy for critical size [16, 18]. We use this proxy for critical size/weight in this study."

2) The authors state: '...in *Manduca sexta* - a Lepidopteran holometabolous insect, larvae between 4th instar and 5th instar stage, whose head capsule size was greater than 5.4 mm were able to successfully pupate else undergo one more molt to 6th instar thus monitor their size by the growth of head capsule in proportion to body size.'

This is a slight overstatement of the conclusions of the cited paper. Nijhout observed a correlation between head capsule size and probability of pupating, and hypothesized that the larvae are monitoring their size by growth of the head capsule (which only grows between instars). However, this was a hypothesis. I am aware of no subsequent study that has explored this hypothesis further. I would therefore modify this sentence to say that the data 'suggest' that larvae monitor their size by the growth of head capsule.

Author's Response to Decision Letter for (RSOS-191910.R1)

See Appendix C.

Decision letter (RSOS-191910.R2)

21-May-2020

Dear Dr SHAKARAD,

It is a pleasure to accept your manuscript entitled "Evolution of reduced minimum critical size as a response to selection for rapid pre-adult development in *Drosophila melanogaster*." in its current form for publication in Royal Society Open Science.

on behalf of Professor Laura Johnston (Associate Editor) and Kevin Padian (Subject Editor)
openscience@royalsociety.org

Appendix A**ROYAL SOCIETY
OPEN SCIENCE****Higher instantaneous growth during second larval instar
leads to lower critical size in *Drosophila melanogaster*
populations**

Journal:	Royal Society Open Science
Manuscript ID	RSOS-191910
Article Type:	Research
Date Submitted by the Author:	19-Dec-2019
Complete List of Authors:	Sharma, Khushboo; University of Delhi, Zoology Mishra, Nalini; University of Delhi, Department of Zoology, Evolutionary Biology Lab SHAKARAD, MALLIKARJUN; University of Delhi, Department of Zoology
Subject:	evolution < BIOLOGY, developmental biology < BIOLOGY
Keywords:	critical size, larval growth, adult body size, adaptive-bailout, accelerated pre-adult development
Subject Category:	Biology (whole organism)

Author-supplied statements

Relevant information will appear here if provided.

Ethics

Does your article include research that required ethical approval or permits?:

This article does not present research with ethical considerations

Statement (if applicable):

CUST_IF_YES_ETHICS :No data available.

Data

It is a condition of publication that data, code and materials supporting your paper are made publicly available. Does your paper present new data?:

Yes

Statement (if applicable):

Data submitted at Dryad can be accessed with the following link

<https://doi.org/10.5061/dryad.k6djh9w32>

Reviewer URL:

<https://datadryad.org/stash/share/4dXJwU3WSx-xFAZVFm15zTrFMNvVj5u9jBMFWiP17h0>

Conflict of interest

I/We declare we have no competing interests

Statement (if applicable):

CUST_STATE_CONFLICT :No data available.

Authors' contributions

This paper has multiple authors and our individual contributions were as below

Statement (if applicable):

MS and KS conceived and designed the study. KS and NM collected data and performed the experiments. KS and MS analysed the data and wrote the manuscript. All authors gave final approval for manuscript submission.

Title

Higher instantaneous growth during second larval instar leads to lower critical size in *Drosophila melanogaster* populations

Khushboo Sharma, Nalini Mishra and Mallikarjun N Shakarad¹
Evolutionary Biology Laboratory, Department of Zoology, University of Delhi, Delhi-110007.

¹Corresponding Author- email id: beelab.ms@gmail.com

36

Background

Holometabolous insect species are characterized by two distinct phases in their life cycle viz., the
pre-adult phase which consist of (i) larval and (ii) pupal stages; and adult phase. During the larval
life, the energy required for metamorphosis from larval to adult tissue and for the early adult life
is accumulated [1, 2, 3]. Further, the duration of the larval stages determines the final adult body
size. Contrary to common belief, unrestricted growth occurs even at the time of moulting due to
the presence of unsclerotized body surface [4]. However, the timing of metamorphosis imposes
restriction on larval duration which directly affects the final adult size and associated life-history
traits [5, 6]. Different mechanisms of final body size assessment exist in insects that are
prerequisite for metamorphosis initiation. In holometabolous and some hemimetabolous insects,
the process of initiation of metamorphosis is dependent on attaining a certain minimum threshold
size called critical size [7-10] beyond which starvation does not alter the time course to
metamorphosis [10, 11-17]. Critical size is similar to minimal viable weight in *Drosophila*
*melanogaster* [10, 11, 14, 18]. Hence, throughout this study we have used the term 'critical
size/weight'.

Critical size being an essential checkpoint during the larval life, acts as a developmental switch
for the irreversible process of metamorphosis [13, 14, 17]. The early phase of larval life consists
of the exponential growth phase that ends in the attainment of critical size while later-post critical
phase is marked by linear growth period on the arithmetic scale [19]. In *Drosophila* sp. the final
adult body size is determined during this post-critical phase. Thus the larval duration is split into
(i) pre-critical duration which is defined as the development time spent in attaining the minimum
size necessary to complete metamorphosis and emerge as an adult [10], and (ii) post-critical
duration, during which additional energy required for maximizing Darwinian fitness is acquired
[1,15]. Once critical size is attained variable size controlling mechanisms operate in different
species before they undergo metamorphosis indicating that these species have unique modes of
determining the body size with critical size at its core [4, 9, 19]. For example, in *Manduca sexta* -
a Lepidopteran holometabolous insect, larvae monitor their size by the relative growth of various
organs in proportion to each other [4]; while in *Oncopeltus fasciatus* -a Hemipteran
hemimetabolous insect, larval growth and its size is estimated by abdominal stretch receptors [9].
Critical size is suggested to evolve in response to environmental conditions. If the environmental
conditions are conducive, then slower growth along with larger body size as a consequence of

69 larger critical size is favoured by natural selection; while under non-conducive environment, faster
growth with the smaller critical size is selected [20]. *Drosophila melanogaster* is known to occupy
ephemeral habitat with limited food and high density and this holometabolous species is under
strong selection for faster pre-adult development [21]. Previously, it has been reported that
*Drosophila* populations under conscious selection for shorter pre-adult duration have reduced
body size [22- 25]. It has been speculated that critical size might reduce if exposed to conscious
selection for accelerated pre-adult development [24]. However, every extant species should have
evolved a species-specific critical size that has been optimised over the course of evolution, as
the critical size is crucial to survival itself. Previous studies have demonstrated evolution of critical
size in *Drosophila melanogaster* populations under conditions of malnutrition [25] and selection
for body size [26], thus exhibiting genetic variability for the trait [11]. For example under direct
artificial selection for change in body size there is reduction of critical size while in another study
populations under nutritional stress leads to smaller critical size. In this study, we test the
hypothesis that selection for faster pre-adult development reduces the critical size in *Drosophila*
*melanogaster* [24].

We used six populations of *D. melanogaster*, of which three were ancestral controls maintained
on a 21-day discrete generation cycle and three were simultaneously selected for faster pre-adult
development and extended reproductive longevity. The control *Drosophila melanogaster*
populations had been through 232 generations of maintenance on 21 days, egg-to-egg discrete
generation cycles while the selected populations had been through 126 generations of
simultaneous selection for faster pre-adult development and indirect selection for extended adult
longevity at the time of initiation of these experiments. We first assessed the pre- and post-critical
duration and critical size in the control and selected populations. Then we evaluated the impact
of non-availability of food on the post-critical larval duration, pupal duration and adult body size in
control and selected populations. Further, we assessed the impact of selection on larval growth
rate. We found that the selected populations have evolved a significantly reduced pre- and post-
critical duration and smaller critical size as a correlated response to selection for faster pre-adult
development. Interestingly, the selected populations have higher growth rate during the second
larval instar suggesting that they might have preponed their growth owing to a very short post-
critical duration.

**Methods**

**a) Fly husbandry**

A total of six *Drosophila melanogaster* populations were used in this study. Of the six populations,
three were ancestral control, maintained on a 21-day egg-to-egg discrete generation cycle. The
other three were simultaneously selected for faster pre-adult development and extended adult
life-span. All the six populations were maintained as outbred populations in Power Scientific Inc.
USA environmental chamber/incubators under standard laboratory conditions (SLC) of 25 ± 1 °C
temperature, $70 \pm 5\%$ RH (Relative Humidity) and 24:0 L:D (Light: Dark) cycle. The pre-adult
stages were reared in glass vials (9.5 cm × 2.3 cm) with 6 mL standard media-SM, (Table 1) and
the adults were reared in plexiglass cages (25 cm × 20 cm × 15 cm). The pre-adults were on a
single meal of SM in glass vials till emergence as adults, while the adults (in plexiglass cages)
were provided fresh SM every alternate day. All population cages were provided with yeast-acetic
acid supplement along with fresh SM 3 days prior to collection of embryos for starting the next
cycle. Each of the control populations was generated in 40 vials, with 50-60 eggs in 6 mL SM per
vial and incubated at SLC for 12 days in vials. All the emerging adults of a given population were
transferred to a clean, sterile pre-labelled population cage with a fresh plate of SM (Figure 1).
Selected populations were derived from corresponding ancestral controls by transferring 60-80
eggs into 6 mL SM vials under SLC. Egg density was kept low so as to avoid larval crowding [27],
and the difference in the egg densities of control and selected populations is marginal and is
unlikely to differentially affect any traits in the two population types. A total of 160 vials per
replicate population were set up. Only the first 15-20 flies emerging from each of 160 vials were
transferred to pre-labelled clean breeding cages through the process of 2 hourly vigil checks. The
initial population size of each of the selected populations was 2400-3200 individuals. In order to
avoid crowding during the adult stage, the emerging adult flies were maintained in two sister
cages, with each cage housing adults from 80 vials. Eggs for initiating the subsequent generation
were collected after 50% adult mortality was noticed in either of the cages, thus ensuring a
breeding population size of ~1600 flies (like control populations) at the time of egg collection for
initiation of next-generation. The eggs from the two sister cages were mixed and redistributed into
160 vials to avoid independent evolutionary trajectories in the two sister cages (Figure 1).

**b) Generation of flies for experiments**

In order to remove the non-genetic parental effects, eggs were collected from both, selected and
control populations, and reared under similar conditions wherein the selection criteria were
relaxed in the selected populations prior to experimentation [22, 28]. Eggs were collected on a
sterile media plate and exact counts of 50 eggs were dispensed into vials with 6 mL of SM. Forty
such vials were maintained per population. Though selected populations were maintained at 60-

136 80 eggs per 6 mL SM in running stock, they are unlikely to experience scramble competition;
especially due to their reduced feeding rates [29]. Further, the marginal difference in the egg
density used for generation of standardised flies and that of running stocks is unlikely to influence
our results. The egg collection from the selected and control populations was staggered by the
developmental time difference to obtain adult flies of the same age. All emerging flies were
transferred to pre-labelled clean and dry population cages with SM plate either on day 10
(selected) or day 12 (control) from the day of egg collection. These flies are referred to as
standardized flies. In the experiments that required large number of embryos, two sister cages of
standardized flies were generated by incubating 80 vials of 50 eggs each, per population.

**c) Larvae collection and fly media**

Prior to the collection of synchronized eggs, standardized fly populations were supplemented with
a generous amount of live-yeast and acetic acid paste for 3 days to boost their egg-laying. After
3 days, they were provided with fresh sterile SM plate for one hour (h) and at the end of 1 h, SM
plate was replaced by uncontaminated non-nutritive agar plate at every one-hour interval for 3

[revised manuscript text omitted]

$$f(x) = \frac{1}{\sigma\sqrt{2\pi}} e^{-\frac{(x_i - \mu)^2}{2\sigma^2}}$$

{Where s represented standard deviation (σ), s^2 represented variance (σ^2), x represented mean
(μ), $x_i = (\mu + \sigma)$ or $(\mu + 2\sigma)$ or $(\mu + 3\sigma)$. $\pi = 3.14$, $e = 2.71$ }

**Results**

**a) Selection for accelerated development leads to the evolution of smaller critical size**

There was significant effect of selection on the critical size ($F_{1,2} = 24.45$, $p = 0.0385$; Figure 2A)
and critical duration ($F_{1,2} = 192.66$, $p = 0.0034$; Figure 2B). The selected populations attained
their critical size at an average wet weight of 1002.66 μg in an average duration of 62.5 h
compared to their ancestral control whose average wet weight was 1308.71 μg attained in 74 h.

A reduction of 23.38% in critical weight was attained with a reduction of 15.31% in critical
developmental duration.

Further, there was significant impact of selection on post-critical developmental duration ($F_{1,2} =$
344.32 , $p = 0.003$; Figure 2C). The developmental duration, post- attainment of critical size was
reduced by 56.8% in the selected populations as compared to their ancestral control populations.

Furthermore, there was no significant effect of the availability of food on the post-critical
developmental duration of larvae in both the selected and control populations ($F_{1,2} = 0.763$, $p =$
0.473 ; Figure 2C). In addition, the reduction in the pupal duration was also non-significant ($F_{1,2} =$
5.960 , $p = 0.135$; Figure 2D) between the selected (86.36 h) and control (89.96 h) populations.

Overall, the egg to adult development time significantly ($F_{1,2} = 363.701$, $p = 0.003$; Figure 2D)
reduced by 17.5% in selected populations compared to their ancestral controls. An average adult
from populations under selection for faster pre-adult development took 188.34 h to eclose from
the egg, while control populations took 228.3 h to eclose.

**b) Selection for accelerated pre-adult development affects larval growth rate at second**
**instar**

The reduction in the critical size associated with a reduction in larval developmental duration of
the selected compared to the control populations could be a correlated response without any
change in the larval growth rate. To address this, the larval growth (measured as wet weight)
trajectories of the two population types were ascertained at every 4 h intervals from the time of
hatching till pupation (Figure 3A). Linear regression analysis of the three larval stages showed no
significant difference in the slope during the L1 (First 24 h, $t = 0.98$; Figure 3A) and L3 stages
(Post 48 h till wandering stage mid-point, $t = 0.16$; Figure 3A; Table 2). However, the slope of the
selected populations was significantly higher than that of their ancestral control during the L2
stage ($t = 3.54$, $p < 0.01$; Figure 3A; Table 2). The increased growth rate was not due to increase
in the feeding rate that was not significantly different between the selected and control populations
($F_{1,2} = 16.14$, $p = 0.057$; Figure 3B).
**c) Impact of selection for accelerated pre-adult development on adult body size and its**
**distribution**

We found a significant reduction ($F_{1,2} = 35.682$, $p = 0.027$; Figure 4A) in the fresh/wet weight of
adults as a function of selection. There was a reduction of 19.13% in the wet weight of an average
fly from the selected populations (689.57 μ g) in comparison to an average fly from control
population (852.73 μ g) when they had access to *ad libitum* food. Further, there was a significant
effect of feeding regimen on the wet weight of the flies ($F_{1,2} = 498.54$, $p = 0.002$; Figure 4A). The
overall reduction in the weight of the flies that emerged after feeding only up to critical duration in
comparison to those that fed till they naturally wondered off to pupate was 51.33%. There was no
selection \times feeding duration interaction effect ($F_{1,2} = 0.031$, $p = 0.877$).

In *D. melanogaster* body size is tightly correlated with development time, thus one would expect
the body size distributions of the selected and control populations to be non-overlapping. In order
to test this hypothesis, we constructed a normal distribution using a normal probability density
function. The normal probability distribution of the adult size (measured as wet weight at
emergence) of the flies from the selected populations was shifted to the left of the normal
probability distribution of control population flies (Figure 4B).
**Discussion**

In *D. melanogaster*, although intricate regulatory hierarchy is suggested to respond to variations

in nutrient availability and ensure uniformity of species-specific final body size [37], selection for
faster pre-adult development have reported a significant reduction in body size [22, 24, 38]. The
reduced adult size concomitantly resulted in reduced lifespan and fecundity [3,8]. However, in an
earlier study, we reported higher fecundity in flies that were significantly smaller [39]. In *D.*
*melanogaster*, it has also been reported that much of the resources that are utilized for survival
during metamorphosis and early adult activities are acquired in the late L3 stage, post attainment
of critical size and are stored in larval fat bodies [1, 2]. However, the post-critical duration in the
populations selected for faster pre-adult development has significantly reduced- suggesting that
these populations are likely to have evolved mechanisms so as to have higher fitness compared
to their ancestral control populations.

Adult body size in insects is an important fitness governing trait that is determined by three factors:
(i) the number of larval instars, (ii) the size increment at each larval moult, and (iii) the size at
which the last larval instar stops feeding and initiates metamorphosis [40]. Many earlier studies
have suggested the critical size to be species-specific [12, 25]. A couple of studies in *Drosophila*
[10, 11] have shown the critical size to be static with respect to nutrition, thus supporting the above
suggestion. However, many studies over the past three decades have shown a reduction in the
adult size of *D. melanogaster* flies in response to lack of nutrition [25], increased growth
temperatures [12] and selection for faster pre-adult development [24, 39]. The decreased adult
size could be due to reduction in the number of larval instars, decrease in the size increment at
each larval moult, and or the size at which the last larval instar stops feeding and initiates
metamorphosis. An additional factor that can alter the final adult size is the duration of each of
the larval stages. A reduction in the duration of 1st and 3rd larval instar contributing to reduction in
final adult size has been reported by Prasad *et al.* [24]. Contrary to many studies [11, 12], Prasad
*et al.* [24] suggested that the critical size in *D. melanogaster* can evolve under selection for faster
pre-adult development. Here we show that populations under selection for faster pre-adult
development have evolved significantly smaller critical size (Figure 2A) supporting the view that
critical size in *D. melanogaster* is polyphenic under different growth conditions [19, 25]. *Drosophila*
*melanogaster* inhabits ephemeral environment where nutritional conditions are deteriorating
continuously thus under the constant pressure of faster development. Larvae exposed to poor
dietary condition since the start of larval life exhibit higher metabolic efficacy and accelerated
development rate through evolution of smaller critical size [25]. *Scathophaga stercoraria* like
*Drosophila melanogaster* adopt adaptive bail out under the condition of continuously deteriorating

environment and accelerates its development along with lower critical size attainment indicating
critical weight to be the target of selection in the yellow dung fly *Scathophaga stercoraria* [41].
The body size is a highly plastic trait, influenced by both genotype as well as the environment.
The plasticity of body size helps the organisms to survive fluctuating food availability, both
quantitatively and qualitatively [42]. A general phyletic trend in the evolution of larger body size
among insects has been through decrease in the number of larval instars accompanied by size
increment at each larval instar [40]. However, adaptive evolutionary processes operate on the
variation present within a population rather than in different populations let alone species, unless
they occupy similar/identical niches. In *M. sexta*, evolution of large body size is accompanied by
an increase in size increment and not an increase in the number of larval instars [43]. Contrary to
the expectation based on phyletic trend and data on *M. sexta*, the reduced adult size in our study
was not accompanied by reduction in size increment at each larval moult (Figure 3). As opposed
to previous study where lines selected for small body size grew slowly [26], populations under
selection for faster pre-adult development demonstrated higher growth rate than control
populations. There was no significant difference in the larval growth rate of the selected
populations and their ancestral control populations during the first as well as the third larval instars
(Figure 3B). Vijendravarma et al. [25] also reported no difference in the growth trajectories of their
selected and control populations till attainment of critical size. However, the growth rate of the
faster developing populations in our study was significantly higher during the second larval instar,
unlike that of Partridge et al. [26] study where the growth rate were lower for the smaller adult size
selections and higher for the larger adult size selections. The increased growth rate during second
larval instar accompanied by reduced critical size could be responsible for the significant reduction
in the post-critical duration of the larvae in our selected populations. The observed differences in
the growth rates in the three studies might be due to the difference in the genotypes of the
populations under study and/or genuinely indicate the dynamic nature of responses due to
different components being the target of selection. The reduction in adult body size could be due
to reduction in the critical size and/or reduction in the post-critical duration. This could be another
form of adaptive-bailout [44] as in the case of *S. stercoraria* in response to food limitation [41, 44].
The populations under selection for faster development might be exhibiting adaptive-bail out [44]
not due to external food limitation but due to internal trigger.

**Conclusion**

Overall, our study provides the experimental evidence for increased larval growth rate specifically
in a developmentally important stage of second instar leading to a reduced critical size and
eventually reduced adult size in *Drosophila melanogaster* populations under selection for faster
development.

**Data availability**

Data is available on Dryad Digital Repository with the following link

<https://doi.org/10.5061/dryad.k6djh9w32> [45]

{Reviewer URL: [https://datadryad.org/stash/share/4dXJwU3WSx-](https://datadryad.org/stash/share/4dXJwU3WSx-xFAZVFm15zTrFMNvVj5u9jBMFWiP17h0)
[xFAZVFm15zTrFMNvVj5u9jBMFWiP17h0](https://datadryad.org/stash/share/4dXJwU3WSx-xFAZVFm15zTrFMNvVj5u9jBMFWiP17h0)}

**Authors' contributions**

MS and KS conceived and designed the study. KS and NM collected data and performed the
experiments. KS and MS analysed the data and wrote the manuscript.

**Competing interests**

Author declares no competing interests.

**Funding**

This work was supported by Council for Scientific and Industrial Research [CSIR-Grant number:
37(1495/11/EMR-II)] for doctoral funding and University of Delhi [R & D Grant, DU: DURC/2014-
2015].

**Acknowledgements**

KS thank Council for Scientific and Industrial Research. NM and MS thank R & D grant of
University of Delhi.

**References**

- 1. Aguila JR, Suszko J, Gibbs AG, Hoshizaki DK. 2007 The role of larval fat cells in adult
*Drosophila melanogaster*. *J. Exp. Biol.* **210**, 956–63.(doi: 10.1242/jeb.001586)

2. Arrese EL, Soulages JL. 2010 Insect Fat Body: Energy, Metabolism, and Regulation.
*Annu. Rev. Entomol.* **55**, 207–25. (doi: 10.1146/annurev-ento-112408-085356)
3. Merkey AB, Wong CK, Hoshizaki DK, Gibbs AG. 2011 Energetics of metamorphosis
in *Drosophila melanogaster*. *J. Insect Physiol.* **57**, 1437–45. (doi:
10.1016/j.jinsphys.2011.07.013)
4. Nijhout HF, Williams CM. 1974 Control of moulting and metamorphosis in the Tobacco
Hornworm, *Manduca sexta* (L.): growth of the last-instar larva and the decision to
pupate. *J. Exp. Biol.* **61**, 481–91.
5. Roff D. 2002 *Life history evolution*. Sunderland, MA: Sinauer Associates.
6. Stearns SC. 1992 *The Evolution of life histories*. Oxford, UK: Oxford University Press.
7. Blakley N, Goodner SR. 1978 Size-dependent timing of metamorphosis in milkweed
bugs (*Oncopeltus*) and its life history implications. *Biol. Bull.* **155**, 499–510. (doi:
10.2307/1540786)
8. Davidowitz G, D'Amico LJ, Nijhout HF. 2003 Critical weight in the development of
insect body size. *Evol. Dev.* **5**, 188–97. (doi:10.1046/j.1525-142X.2003.03026.x)
9. Nijhout HF. 1979 Stretch-induced moulting in *Oncopeltus fasciatus*. *J. Insect Physiol.*
**25**, 277–81. (doi: 10.1016/0022-1910(79)90055-6)
10. Robertson FW. 1963 The ecological genetics of growth in *Drosophila* 6. The genetic
correlation between the duration of larval period and body size in relation to larval diet.
*Genet. Res.* **4**, 74-92. (doi: 10.1017/S001667230000344X)
11. De Moed GH, Kruitwagen CLJJ, De Jong G, Scharloo W. 1999 Critical weight for the
induction of pupariation in *Drosophila melanogaster*: genetic and environmental
variation. *J. Evol. Biol.* **12**, 852–8. (doi: 10.1046/j.1420-9101.1999.00103.x)
12. Ghosh SM, Testa ND, Shingleton AW. 2013 Temperature-size rule is mediated by
thermal plasticity of critical size in *Drosophila melanogaster*. *Proc. R. Soc. B* **280**,
20130174. (doi: 10.1098/rspb.2013.0174)
13. Mirth CK, Riddiford LM. 2007 Size assessment and growth control: how adult size is
determined in insects. *BioEssays* **29**, 344–55. (doi: 10.1002/bies.20552)
14. Mirth C, Truman JW, Riddiford LM. 2005 The role of the prothoracic gland in
determining critical weight for metamorphosis in *Drosophila melanogaster*. *Curr.*
*Biol.* **15**, 1796–807. (doi: 10.1016/j.cub.2005.09.017)

15. Ohhara Y, Kobayashi S, Yamanaka N. 2017 Nutrient-dependent endocycling in
steroidogenic tissue dictates timing of metamorphosis in *Drosophila melanogaster*.
*PLoS Genet.* **13**, 1–21. (doi: 10.1371/journal.pgen.1006583)
16. Stieper BC, Kupershtok M, Driscoll MV, Shingleton AW. 2008 Imaginal discs regulate
developmental timing in *Drosophila melanogaster*. *Dev. Biol.* **321**, 18–26. (doi:
10.1016/j.ydbio.2008.05.556)
17. Suzuki Y, Koyama T, Hiruma K, Riddiford LM, Truman JW. 2013 A molt timer is
involved in the metamorphic molt in *Manduca sexta* larvae. *Proc. Natl Acad. Sci.* **110**,
12518–25. (doi: 10.1073/pnas.1311405110)
18. Rewitz KF, Yamanaka N, O'Connor MB. 2013 Developmental checkpoints and
feedback circuits time insect maturation. *Curr. Top. Dev. Biol.* 1–33.
(doi:10.1016/B978-0-12-385979-2.00001-0)
19. Royes WV, Robertson FW. 1964 The nutritional requirements and growth relations of
different species of *Drosophila*. *J. Exp. Zool.* **156**, 105–35. (doi:
10.1002/jez.14015601081)
20. Wilbur HM, Collins JP. 1973 Ecological aspects of amphibian metamorphosis:
Nonnormal distributions of competitive ability reflect selection for facultative
metamorphosis. *Science* **182**, 1305–14. (doi: 10.1126/science.182.4119.1305)
21. Santos M, Borash DJ, Joshi A, Bounlutay N, Mueller LD. 1997 Density-dependent
natural selection in *Drosophila*: evolution of growth rate and body size. *Evolution* **51**,
420–32. (doi: 10.1111/j.1558-5646.1997.tb02429.x)
22. Chippindale AK, Alipaz JA, Chen H-W, Rose MR. 1997 Experimental evolution of
accelerated development in *Drosophila*. 1. Developmental speed and larval survival.
*Evolution* **51**, 1536. (doi: 10.2307/2411206)
23. Nunney L. 1996 The response to selection for fast larval development in *Drosophila*
*melanogaster* and its effect on adult weight: an example of a fitness trade-off. *Evolution*
**50**, 1193–204. (doi: 10.1111/j.1558-5646.1996.tb02360.x)
24. Prasad NG, Shakarad M, Anitha D, Rajamani M, Joshi A. 2001 Correlated responses
to selection for faster development and early reproduction in *Drosophila*: the evolution
of larval traits. *Evolution* **55**, 1363–72. (doi: 10.1111/j.0014-3820.2001.tb00658.x)
25. Vijendravarma RK, Narasimha S, Kawecki TJ. 2011 Chronic malnutrition favours
smaller critical size for metamorphosis initiation in *Drosophila melanogaster*. *J. Evol.*
*Biol.* **25**, 288–92. (doi: 10.1111/j.1420-9101.2011.02419.x)

26. Partridge L, Langelan R, Fowler K, Zwaan B, French V. 1999 Correlated responses to
selection on body size in *Drosophila melanogaster*. *Genet Res.* **74**, 43–54.
(doi:10.1017/S0016672399003778)
27. Joshi A, Mueller LD. 1996 Density-dependent natural selection in *Drosophila*: Trade-
offs between larval food acquisition and utilization. *Evol. Ecol.* **10**, 463–74. (doi:
10.1007/BF01237879)
28. Prasad NG, Shakarad M, Gohil VM, Sheeba V, Rajamani M, Joshi A. 2000 Evolution
of reduced pre-adult viability and larval growth rate in laboratory populations of
*Drosophila melanogaster* selected for shorter development time. *Genet. Res.* **76**, 249–
59. (doi: 10.1017/S0016672300004754)
29. Rajamani M, Raghavendra N, Prasad NG, Archana N, Joshi A, Shakarad M. 2006
Reduced larval feeding rate is a strong evolutionary correlate of rapid development in
*Drosophila melanogaster*. *J. Genet.* **85**, 209–12. (doi: 10.1007/BF02935333)
30. Chandrashekara KT, Shakarad MN. 2011 Aloe vera or resveratrol supplementation in
larval diet delays adult aging in the fruit fly, *Drosophila melanogaster*. *J. Gerontol. A*
*Biol. Sci. Med. Sci.* **66**, 965–71. (doi: 10.1093/gerona/glr103)
31. Aditi K, Shakarad MN, Agrawal N. 2016 Altered lipid metabolism in *Drosophila* model
of Huntington's disease. *Sci. Rep.* **6**, 31411 (doi: 10.1038/srep31411)
32. Edgecomb RS, Harth CE, Schneiderman AM. 1994 Regulation of feeding behavior in
adult *Drosophila melanogaster* varies with feeding regime and nutritional state. *J. Exp.*
*Biol.* **197**, 215–35.
33. IBM Corp. 2013 IBM SPSS Statistics for Windows, Version 22.0. Armonk, NY: IBM
Corp.
34. Alpatov WW. 1929 Growth and variation of the larvae of *Drosophila melanogaster*. *J.*
*Exp. Zool.* **52**, 407–37. (doi: 10.1002/jez.1400520303)
35. Sokal RR, Rohlf FJ. 1981 *Biometry: The principles and practice of statistics in*
*biological research* (2nd Ed). San Francisco: W. H. Freeman and Co.
36. Sokal RR, Rohlf FJ. 1995 *Biometry: The principles and practice of statistics in*
*biological research* (3rd Ed). New York: W. H. Freeman and Co.
37. Callier V, Nijhout HF. 2013 Body size determination in insects: a review and synthesis
of size- and brain-dependent and independent mechanisms. *Biol. Rev.* **88**, 944–54.
(doi: 10.1111/brv.12033)

38. Yadav P, Sharma VK. 2013 Circadian clocks of faster developing fruit fly populations
also age faster. *Biogerontology* **15**, 33–45. (doi: 10.1007/s10522-013-9467-y)
39. Handa J, Chandrashekara KT, Kashyap K, Sageena G, Shakarad MN. 2014 Gender
based disruptive selection maintains body size polymorphism in *Drosophila*
*melanogaster*. *J. Biosci.* **39**, 609–20. (doi: 10.1007/s12038-014-9452-x)
40. Nijhout H, Davidowitz G, Roff D. 2006 A quantitative analysis of the mechanism that
controls body size in *Manduca sexta*. *J. Biol.* **5**, 16. (doi: 10.1186/jbiol43)
41. Rohner PT, Blanckenhorn WU, Schäfer MA. 2017 Critical weight mediates sex-
specific body size plasticity and sexual dimorphism in the yellow dung fly *Scathophaga*
*stercoraria* (Diptera: Scathophagidae). *Evol. Dev.* **19**, 147–56. (doi:
10.1111/ede.12223)
42. Parker J, Johnston LA. 2006 The proximate determinants of insect size. *J. Biol.* **5**, 15.
(doi: 10.1186/jbiol47)
43. Nijhout HF. 1994 *Insect Hormones*. Princeton: Princeton University Press.
44. Blanckenhorn WU. 1999 Different growth responses to temperature and resource
limitation in three fly species with similar life histories. *Evol. Ecol.* **13**. (doi:
10.1023/A:1006741222586)
45. Sharma K, Mishra N, Shakarad M 2019 Data from: Higher instantaneous growth during
second larval instar leads to lower critical size in *Drosophila melanogaster* populations.

Figure 1 Schematics of control and selected population life cycle. The control population was maintained on 21 days of egg-to-egg discrete life cycle, one cage per population for adult stages. The selected population is under conscious selection for accelerated development and indirect selection for extended longevity. The selected population is maintained in twin cages per replicate to avoid over-crowding during adult stage. Both control and selected populations are maintained under SLC in Power Scientific USA incubators.

246x185mm (96 x 96 DPI)

Figure 2 (a) Average critical size, (b) Average pre-critical duration, (c) Average post-critical duration, and (d) Average development duration from egg to adult eclosion. The black bar represents selected population and orange stands for control population. Grey bar, pink bar represents food availability until critical size time point only and ad libitum food during larval duration in both populations respectively. Different shades of green bars from light to dark stands for egg, pre-critical, post-critical and pupal duration respectively.

241x192mm (96 x 96 DPI)

Figure 3 (a) Larval growth rate in terms of weight gain in control population (orange) is up to 108 h prior to pupation and in the selected population (black) is maximal up to 76 h and then it undergoes metamorphosis. (b) Larval feeding rates at L1, L2 and L3 stages.

246x115mm (96 x 96 DPI)

Figure 4 (a) Adult body weight (μg), Grey- when larvae were on non-nutritive agar post attaining critical weight, and Pink- when larvae fed till they naturally stopped feeding and wandered to pupate. (b) Adult weight probability density functions of selected (black) and control (orange) populations.

246x115mm (96 x 96 DPI)

Diet composition (1 L)	Standard Media (SM)	Liquid Standard Media (LSM)
Water	1180 mL	1180 mL
Banana	205 g	205 g
Jaggery	35 g	35 g
Barley flour	25 g	25 g
Yeast	36 g	36 g
Methyl paraben	2.4 g	2.4 g
Ethanol	45 mL	45 mL
Agar-agar	12.5 g	Zero

Table 1 **Diet composition:** Standard media and Liquid standard media differ only with respect to agar-agar composition.

Larval duration- β values	Control populations	Selected populations	t- values
L1- Zero to 24h post hatching	1.66	2.17	0.97
L2- 24h to 48h post hatching	8.62	17.21	3.54
L3- 48h to wandering stage mid- point*	49.77	29.49	0.16

*For selected populations, as L3 is of small duration thus midpoint value is considered for the analysis. L1, L2 and L3 stand for first, second and third larval stages.

Table 2 **Regression table:** β value of control and selected populations and respective t values at different larval stages.

Appendix B

Reviewer 1

- 1) The authors suggest that the terms minimal viable weight (MVW) and critical size (CS) are interchangeable, while several researchers have demonstrated that they are distinct phenomena. The authors correctly define CS as the size above which starvation does not alter the time course to metamorphosis. The CS is therefore a physiological switch in the effect of nutrition on the hormone cascade that ends on metamorphosis, and the molecular genetic mechanisms underlying the switch have been well elucidated. The MVW for pupariation (MVW_{pupariation}) is the minimum weight at which >50% of starved larvae pupariate, while the MVW for eclosion (MVW_{eclosion}) is the minimum weight at which >50% of starved larvae eclose as adults. Both of these therefore reflect a larva's nutritional status and its ability to mobilize stored resources and survive to different points in development upon starvation. While the three phenomena are correlated, they are not the same. Generally MVW_{pupariation} < MVW_{eclosion} < CS (see Stieper et al 2008 and Hironaka et al 2019 for more on this). Further, only *Drosophila* researchers confound CS and MVW: in *M. sexta* for example, CS is the time at which starvation no longer affects the time course to metamorphosis.

While I think that it is fine for the authors to use MVW_{eclosion} as a proxy for CS (as it has been used in various publications, e.g. Mirth et al 2014, Hironaka et al 2019), it adds confusion if the distinction is not clearly stated in the text. Further, it is conceivable that an observed reduction in MVW_{eclosion} may not be accompanied by a reduction in CS *sensu stricto*. Problematically, while the CS phenomenon is well understood at a physiological level, the MVW phenomenon is not, although it likely involves the initiation of ecdysteroidgenesis by factors other than nutrition, through a bail-out response or a 'leaky' prothoracic gland (see Nijhout et al 2014 WIRE Dev Bio for more details). Thus, while the observed response to selection may indeed reflect changes in the mechanisms that initiate ecdysteroidgenesis at attainment of CS, it is also possible that they reflect changes in the ability to initiate a bail-out response, although these mechanisms remain unknown. Without distinguishing between CS and MVW, the distinction is lost.

Reply:

L48-50: We agree with the reviewer's reasoning and accordingly made the necessary changes in the text. We had used the term critical size (CS) and minimum viable weight

(MVW) interchangeably as in the manuscript (L48-50) as it is widely used in *Drosophila* community. We have now modified the terminology to Minimum viable weight for eclosion instead of MVW and cited Steiper et al., 2008 and Hironaka et al., 2019.

2) L64 “larvae monitor their size by the relative growth of various organs in proportion to each other [4]”. The cited paper does not support the statement (indeed I do not know of any paper that does).

Reply:

There was an error in citing the research article. We have now corrected the error and cited appropriate reference (Nijhout 1975- A Threshold Size for Metamorphosis in the Tobacco Hornworm, *Manduca sexta* (L.). Accordingly, the correction now reads (L62-65) ‘...in *Manduca sexta* - a Lepidopteran holometabolous insect, larvae between 4th instar and 5th instar stage, whose head capsule size was greater than 5.4 mm were able to successfully pupate else undergo one more molt to 6th instar thus monitor their size by the growth of head capsule in proportion to body size.’

3) L67” Critical size is suggested to evolve in response to environmental conditions. If the environmental conditions are conducive, then slower growth along with larger body size as a consequence of larger critical size is favoured by natural selection; while under non-conducive environment, faster growth with the smaller critical size is selected”. This statement has unclear logic. All traits evolve by natural selection in response to ‘environmental conditions [20]’. The question is what is the selective pressure and what is it targeting. Does low nutrition select for smaller body size or accelerated growth? And does it target critical size directly, or is critical size a consequence of selection on body size/growth rate? See Hironaka et al 2019 for more details on how selection may act to change critical size. Further, the citation is for amphibians and not holometabolous insects.

Reply:

We agree with the reviewer in that ‘all traits in response to environmental conditions’. We have now modified the section starting with L67 to read as ‘Critical size is suggested to evolve in response to environmental conditions. For example in *Drosophila* genus, large sized species like *D. repleta* have higher critical size- which is larger than the final larval

size of small sized *D. willistoni*. Critical size change, thus, can be one of the drivers of adult body size evolution (Hironaka et al., 2019) [18].’

4) L80. Please provide citations for the selection experiments.

Reply:

L79 In accordance with the text we cited appropriate research papers- Vijendravarma et al., 2011 [25] and Partridge et al., 1999 [26].

5) L90. Here you correctly argue that the change in critical size is a correlated response to selection for accelerated development. However, in the title you state “Higher instantaneous growth during second larval instar leads to lower critical size in *Drosophila melanogaster* populations”, which is not supported by your data, since both growth rate and change in critical size are a correlated response to changes in developmental time. The title should be changed to reflect this.

Reply:

We thank the reviewer for the kind appreciation. As per both the reviewers’ suggestion, we have adopted the title suggested by 2nd Reviewer and the modified title is:

“Evolution of reduced minimum critical size as a response to selection for rapid pre-adult development in *Drosophila melanogaster*.”

6) In the methods you describe the selection regime, and provide a helpful figure. However, you do not explicitly state how many generations you have maintained selection apart from in the introduction. You should state this in the M&M section also.

Reply:

We have added the generation number in Materials and methods section in L146-149.

7) L130. This section is a little difficult to follow. Problematically you use the term ‘standardized flies’ (L138) before you define it (L143).

Reply:

As per the reviewer's suggestion we have reorganized the text and explicitly defined the 'standardized flies' from L129 to L136 prior to its use.

8) Minor point: The sampling method for calculating MVW/CS is rather complex. A figure would be helpful to show how larvae were sampled at each time point.

Reply:

We have added a figure (Figure 1c) for the larval sampling method as suggested by the reviewer.

9) L249. The authors report the MVW/CS as 1002.66 μg which is likely much more precise than the microbalance that they are using. It would be useful to adjust the number of significant figures to reflect the precision of the microbalance.

Reply:

We agree with the reviewer that we would not be able to accurately measure any biological parameter. However, the precision of microbalance (Citizen- CM 11) used in our experiments is up to 5 decimal places but we had rounded off the data values to two decimals places only.

10) L256 "Furthermore, there was no significant effect of the availability of food on the post-critical developmental duration of larvae in both the selected and control populations" This is an interesting result since many studies have shown that starvation at CS accelerates pupariation in *Drosophila* (e.g. Steiper et al 2009). Does this reflect the relatively imprecise method used to calculate CS/MVW? The challenge with the methodology used is that even though eggs were collected in a 1h cohort, developmental stage can vary considerably by the time larvae get to the 3rd instar (Ashburner discusses this in his book). Thus the average mass of a cohort of larvae at 70h AEL may hide considerable variation in size. Further, the authors only calculated MVW/CS at a resolution of every 2 hours. Other methods to calculate MVW/CS (e.g. using logistic regression, Steiper et al 2009, Hironaka et al 2019) appear to be more precise and may have been able to detect the acceleration in developmental time upon starvation at CS/MVW.

Reply:

We agree that average mass would mask the variability. However, we do not agree with the reviewer that our method is imprecise. Measurements at every two hour interval is based upon a pilot run done prior to assay initiation (replicated thrice). It is humanly not possible to proceed with the experiment at less than two hour interval. As handling of larvae itself takes enormous time thus prolonging the process. This would have resulted in greater variability due to ever changing larval growth. At every time point, we handled a total of 600 larvae (300 larvae per selection line) for weighing and incubation at SLC as per the protocol used.

Further, we agree with the reviewer that no experimental design can remove inter-individual variability, at best one can attempt to minimize it and that is what we have attempted achieve through our experimental protocol.

11) L305 “However, the post-critical duration in the populations selected for faster pre-adult development has significantly reduced- suggesting that these populations are likely to have evolved mechanisms so as to have higher fitness compared to their ancestral control populations.” I do not follow the logic of this argument. Why would this lead to higher fitness? Does the author mean higher fitness under the specified selection pressure?

Reply:

We have modified the statement to read ‘...suggesting that these populations are likely to have evolved mechanisms such as, energy acquisition, storage and utilization during the adult stage due to long adult life owing to the selection protocol, and thus have fitness comparable to their ancestral control populations (Handa et al. 2014) [40]. Incidentally, there was no fitness cost in terms of viability during the pre-adult stages (Supplementary figure 1) suggesting that the long adult life-span seems to have mitigated the viability cost during the pre-adult stage of our selected (FLJ) populations, unlike that reported by Prasad et al. [24] in their FEJ populations selected for faster pre-adult development and early reproduction (L314-323).

12) L312 “A couple of studies in *Drosophila* [10, 11] have shown the critical size to be static with respect to nutrition, thus supporting the above suggestion. However, many studies over the past three decades have shown a reduction in the adult size of *D. melanogaster* flies in response to lack of nutrition [25], increased growth temperatures [12]

and selection for faster pre-adult development [24, 39].” This is rather misleading since the authors confound phenotypic plasticity with adaptation. The cited studies [1-,11] suggest that CS/MVW does not vary within a genotype in response to nutrition; that is, it is not nutritionally plastic. This does not mean that there is no genetic variation for CS/MVW nor that it cannot evolve.

Reply:

We have removed all citations that were dealing with phenotypic plasticity and restricted our citations to selection studies throughout the discussion section.

13) L323 “Here we show that populations under selection for faster pre-adult development have evolved significantly smaller critical size (Figure 2A) supporting the view that critical size in *D. melanogaster* is polyphenic under different growth conditions [19, 25]”. Again, the authors confound plasticity with adaptation. Polyphenism is the response of a phenotype to environmental conditions (strictly speaking two or more discrete phenotypes, e.g. castes of bees). Their data support the hypothesis that CS/MVW is genetically variable and can respond to selection, not that it is phenotypically plastic.

Reply:

We have restricted our discussion section to include studies pertaining to adaptation only and modified the text accordingly.

14) I encourage the authors to revise their figures. The colors are rather difficult to follow (a key in the figure would help). Also, it is increasingly standard to include all the data points in box plots and bar charts, to give the reader an idea of the distribution of data. Finally, the authors must specify what the error bars are and provide sample sizes for all statistical tests.

Reply: We have revised our figures and amended them as per both reviewer’s suggestions.

15) Statistical Analysis. Details of the statistical analyses need to be included. What statistical models were used to analyze the data? The authors used a GLM but did not

specify the link function or probability distribution. How were the regression slopes compared? Why fit a linear regression when your state clearly in the introduction that growth prior to CS/MVW is exponential? Did you test for homoscedasticity?

Reply:

We included the statistical analysis details as per both reviewers' comments.

We fitted linear regression and slopes were compared using t test (L243-247).

16) L289 "In *D. melanogaster* body size is tightly correlated with development time, thus one would expect the body size distributions of the selected and control populations to be non-overlapping". Why would you expect it to be non-overlapping? Different, possible, but not necessarily non-overlapping. Also, I am not sure why the authors examined the impact of selection on body size distribution. Why not just apply a standard two-sample t-test to see whether body size is different?

Reply:

L297: We have provided the actual data (figure 4a). Since we did not weigh individual flies, we tried to recapture the population distribution based on theoretical model so as to have a near realistic picture of the distribution of adult body sizes in the two types of populations. The body size distributions of the two populations were non-overlapping (figure 4b). These results are similar to those reported based on the actual wing measurements reported by our laboratory (Handa et al., 2014) [40]. This will also provide us a view how selection for faster pre-adult development actually affect the body weight as there is no difference in body size at critical size (figure 2c and 4a) implying the significance of post-critical duration in determining the final adult body size (Hironaka et al., 2019) [18].

Reviewer: 2

This is a nice piece of work. The experimental design is good and statistics are appropriate. The issue is one of interest, especially within the fly experimental evolution community. My main concerns are with the way the Introduction and Discussion are framed, which should be improved in a revision. I am attaching an annotated copy of the manuscript with detailed comments to this end. Since I choose not to be anonymous, I urge the authors to feel free to contact me if they have any doubts when they undertake a revision.

Reply:

We are grateful to the reviewer Prof. Amitabh Joshi for his appreciation of our work.

- 1) It is not at all clear that the lower critical size is BECAUSE of the higher growth rate during the 2nd instar. Indeed, one could argue that faster growth could permit a faster attainment of the same critical size, thereby not requiring a reduction in critical size while achieving a reduction in critical feeding time, in the context of selection for reduced egg to adult development time. Why not title it something like "Evolution of reduced minimum critical size as a response to selection for rapid pre-adult development in *Drosophila*" and frame the story in terms of "faster dev can be achieved by faster growth, leading to attaining minimum size faster, and/or by reducing the minimum size"

Reply:

We are thankful to the reviewer for suggesting a more appropriate title. Title of the manuscript has been changed as per both the reviewers' suggestion.

The new title is "Evolution of reduced minimum critical size as a response to selection for rapid pre-adult development in *Drosophila melanogaster*".

- 2) Do you mean replicate populations? If so, were they treated as random blocks (for ancestry/handling) or as random nested replicate populations within selection regimes?

Reply:

Yes, we mean replicate populations and accordingly were treated as random blocks (for ancestry/handling) as in Prasad et al., 2001 [24], Rajamani et al., 2006 [28].

- 3) Not clear why you did this as opposed to simply showing the actual distribution of measured body sizes.

Reply:

We have provided the actual data (Figure 4a). Since we did not weigh individual flies, we tried to recapture the population distribution based on theoretical model so as to have a near realistic picture of the distribution of adult body sizes in the two types of populations. The body size distributions of the two populations were non-overlapping (Figure 4b). These results are similar to those reported based on the actual wing measurements reported by our laboratory (Handa et al., 2014) [40].

- 4) (L260) This actually suggests an overall reduction in average rate of weight gain during the 1st and 2nd instars, since a lot of that critical size weight is lost for a disproportionately smaller reduction in time taken to critical weight. This needs to be addressed.

Reply: We have discussed this elaborately in the discussion (L 350-363). The higher growth rate could possibly be another reason for the lack of pre-adult viability cost in our selected populations. The higher pre-adult mortality in Prasad *et al.* [24] could perhaps be due to lower growth rate in their selected (FEJ) populations. Further, a reduction of 23.38% in critical weight with a reduction of only 15.31% in critical developmental duration suggests that the control (JB) populations might be gaining disproportionately higher weight during the last 12 hours prior to reaching critical size (Supplementary table 1). Unlike the study of Prasad *et al.* where selection for faster development leads to reduction in pupal duration [24], we found no significant difference in the pupal duration of our selected (FLJ) populations (figure 2d). The differences in the results of the two studies despite the ancestral populations being the same, could be due to the differences in the selection pressure in the adult phase. The FEJ populations [24] are under pressure to maximize their fitness by day 3 post emergence while the selected (FLJ) populations used in this study have long adult life. The importance of such cross-life-stage effects in life history evolution have been reviewed in Prasad and Joshi [43].

- 5) Since the selected populations have retained about 45% of their post-critical duration, a time when weight gain rises quite rapidly, it is not clear what is gained by increasing the growth rate in the 2nd instar. Had the selected populations basically evolved to pupate after attaining

critical size, the pressure to increase 1st or 2nd instar growth rate would be more intuitively understandable. This should be taken up in the Discussion.

Reply:

Perhaps the increased growth rate is helping in putting on more weight and buffering them from paying pre-adult viability cost. We have discussed this in detail (L350-363). Besides, the 45% retention in post-critical duration may not necessarily be sufficient enough to build up the required energy reserves.

- 6) No reduction in pupal duration is very different from the obs in the FEJ populations of Prasad et al 2001, which is referenced, This needs to be addressed in the Discussion.

Reply:

We have discussed the differences in the results of the two studies in detail (L356-363). It now reads 'Unlike the study of Prasad *et al.* where selection for faster development leads to reduction in pupal duration [24], we found no significant difference in the pupal duration of our selected (FLJ) populations (Figure 2d). The differences in the results of the two studies despite the ancestral populations being the same, could be due to the differences in the selection pressure in the adult phase. The FEJ populations [24] are under pressure to maximize their fitness by day 3 post emergence while our selected (FLJ) populations have long adult life. The importance of such cross-life-stage effects in life history evolution have been reviewed in Prasad and Joshi [43].'

- 7) Feeding rate (sclerite retractions per min) in the FEJ and FLJ populations used in earlier studies was reduced, relative to controls. Here, of course, you are measuring something different i.e. food ingestion rate but, nevertheless, it would be good to discuss this discrepancy between previous studies and the present one. One speculation I have is that 2 h is long enough for the gut to get fully loaded with the dyed food, and this allows slower feeding selected population larvae to "catch up". Some discussion of this is needed.

Reply:

L212-233, L282-284 & L370-377- We agree with the observation and interpretation of Prof. Joshi with respect to the feeding rate discrepancies in the two studies and modified the terminology to 'larval food ingestion assay' and discussed the results accordingly.

- 8) You should explain why this round-about method is used, as opposed to just showing the reader the body size distributions in the actual data.

Reply:

We have provided the actual data (figure 4a). Since we did not weigh individual flies, we tried to recapture the population distribution based on theoretical model so as to have a near realistic picture of the distribution of adult body sizes in the two types of populations. The body size distributions of the two populations were non-overlapping. These results are similar to those reported based on the actual wing measurements reported by our laboratory (Handa et al., 2014) [40].

- 9) Unclear what you are trying to convey here. Is it the idea that these selected populations having a long adult life before egg-collection have time to feed and put on lipids and hence can sacrifice post-critical feeding without fitness consequences? Incidentally, how is the pre-adult mortality across the selected and control populations?

Reply:

Yes, we meant the non-significant fitness difference despite having lower weight at eclosion (Handa et al., 2014) [40]. We have clearly discussed it in the revised version of the manuscript. For pre-adult mortality, we found non-significant difference in the survival of pre-adult stages till adult eclosion ($F_{1,2} = 4.75$, $p = 0.161$ for larva to adult eclosion up to critical sized fed larva then eclosed as adults; $F_{1,2} = 2.31$, $p = 0.268$ for larva fed up to natural pupation time and eclosed as normal sized adults). We have added the data as supplementary figure (Supplementary figure 1) and we have discussed the implications of these results too (L319-323).

- 10) To my mind, this Discussion is a bit dissipated. Changes in larval instar number would seem largely irrelevant to *Drosophila*. It would be better to organize the discussion of each major Result you have: critical size, critical duration, feeding rate, body size around a comparison of previous work (FEJs) and what we might predict an optimal pre-adult life-history for your selected lines to be.

Reply:

In the revised manuscript we have restricted discussion on interpreting our results and relevant literature and made it more precise.

11) Not clear why you think that critical size reduction is because of increased 2nd instar growth rate. The data do not address any causal link between the two and, on the face of it, they would appear to be two independent results in response to the selection.

Reply:

Again, as per both reviewers' kind suggestions we have modified the conclusion and title of the manuscript.

Appendix C

Reviewer: 1

Comments to the Author(s)

1) The authors state: “In holometabolous and some hemimetabolous insects, the process of initiation of metamorphosis is dependent on attaining a certain minimum threshold size called critical size [7-10] beyond which starvation does not alter the time course to metamorphosis [10, 11-17]. Critical size, also called minimum critical size, is similar to minimal viable weight for eclosion in *Drosophila melanogaster* [16, 18]. Hence, throughout this study we have used the term ‘critical size/weight’.”

While I appreciate the authors clarifying the difference between MVW [eclosion] and critical size, this change rather misses the point. The authors define the critical size as the point at which starvation does not alter the time course to metamorphosis, but do not measure this. Rather they measure the size at which 50% of starved larvae survive to eclosion, which is the MVW [eclosion]. It is imprecise to say that MVW [eclosion] is ‘similar’ to critical size/weight. Rather *Drosophilists* use MVW [eclosion] as a proxy for critical size weight, which is what the authors do in their study. Thus the penultimate sentence of this paragraph should read something like: “In *Drosophila*, the size at which 50% of starved larvae successfully eclose as adults (the minimal viable weight for eclosion) is used as a proxy for critical size [16, 18]. We use this proxy for critical size/weight in this study.”

Reply:

We agree with the reviewer’s reasoning and accepted the text suggested by the Reviewer and have replaced the sentences in L48-50. The penultimate sentence of this paragraph should read something like: “In *Drosophila*, the size at which 50% of starved larvae successfully eclose as adults (the minimal viable weight for eclosion) is used as a proxy for critical size [16, 18]. We use this proxy for critical size/weight in this study.”

2) The authors state: ‘...in *Manduca sexta* - a Lepidopteran holometabolous insect, larvae between 4th instar and 5th instar stage, whose head capsule size was greater than 5.4 mm were able to successfully pupate else undergo one more molt to 6th instar thus monitor their size by the growth of head capsule in proportion to body size.’

This is a slight overstatement of the conclusions of the cited paper. Nijhout observed a correlation between head capsule size and probability of pupating, and hypothesized that the larvae are monitoring their size by growth of the head capsule (which only grows between instars). However, this was a hypothesis. I am aware of no subsequent study that has explored this hypothesis further. I would therefore modify this sentence to say that the data ‘suggest’ that larvae monitor their size by the growth of head capsule.

Reply:

We have changed the sentence in L64-65 as suggested by the Reviewer. The sentence now reads as 'thus the data suggest that larvae monitor their size by the growth of head capsule [20];'